# Prolonging lung cancer response to EGFR inhibition by targeting the selective advantage of resistant cells

Lisa Brunet[1,2], David Alexandre [1,2,17], Jiyoung Lee[1,2,17], Maria del Mar Blanquer-Rosselló [1,2], David Bracquemond[3], Alexis Guernet[1,2], Houssein Chhouri[1,2], Mathilde Goupil[1,2], Zoulika Kherrouche[4], Arnaud Arabo [5], Maicol Mancini [3], Dorthe Cartier[1,2], Shen Yao[6], David Godefroy[1,2], Julie Dehedin[1,2], Jian-Rong Li[7,8], Céline Duparc [1,2], Philippe Jamme[4], Audrey Vinchent[4], Caroline Bérard[9], David Tulasne[4], Sabrina Arena [10,11], Alberto Bardelli [10,12], Chao Cheng [7,8], Byoung Chul Cho [13], Olivier Wurtz[14], Cédric Coulouarn [15], Antonio Maraver [3], Stuart A. Aaronson [6], Alexis B. Cortot[4,16], Youssef Anouar [1,2] & Luca Grumolato [1,2] ✉

Non-small cell lung cancers (NSCLCs) treated with tyrosine kinase inhibitors (TKIs) of the epidermal growth factor receptor (EGFR) almost invariably relapse in the long term, due to the emergence of subpopulations of resistant cells. Through a DNA barcoding approach, we show that the clinically approved drug sorafenib specifically abolishes the selective advantage of EGFR-TKI-resistant cells, while preserving the response of EGFR-TKI-sensitive cells. Sorafenib is active against multiple mechanisms of resistance/tolerance to EGFR-TKIs and its effects depend on early inhibition of MAPK-interacting kinase (MKNK) activity and signal transducer and activator of transcription 3 (STAT3) phosphorylation, and later down-regulation of MCL1 and EGFR. Using different xenograft and allograft models, we show that the sorafenib-EGFR-TKI combination can delay tumor growth and promote the recruitment of inflammatory cells. Together, our findings indicate that sorafenib can prolong the response to EGFR-TKIs by targeting NSCLC capacity to adapt to treatment through the emergence of resistant cells.

Clonal driver mutations are a target of choice for cancer therapy. One of the first and more successful examples of this type of treatment is represented by EGFR-TKIs, including first-generation gefitinib and erlotinib, and third-generation osimertinib, which are used as first-line treatment for advanced *EGFR*-mutant NSCLCs[1]. Despite objective response rates that can exceed 80%[2,3], EGFR-TKIs almost invariably fail in the long term because of the development of resistance[4]. Current guidelines are based on a sequential scheme, in which EGFR-TKIs are administered as single agents until progression, followed by treatment

with chemotherapy or, when an actionable mechanism of resistance is identified, combinations with another targeted drug[5,6].

In recent years, a better understanding of how cancer can adapt to treatment and become insensitive has suggested that, instead of waiting for tumor relapse, therapies could be designed to anticipate and delay the onset of resistance, thus prolonging EGFR-TKI response. Tumors are formed by an intricate combination of heterogeneous subclonal populations that evolve in the presence of a selective pressure, such as therapeutic intervention[7]. As for other targeted drugs,

EGFR-TKI treatment provokes over time the emergence of resistant clones, which can be present before the therapy begins or originate de novo from pools of tolerant/persister cells[8–11], eventually resulting in tumor relapse. Different drug combinations have been proposed to interfere with these processes and inhibit the acquisition of resistance[12]. Some of them are meant to enhance inhibition of EGFR signaling by targeting other components of the pathway, using compounds that act synergistically with EGFR-TKIs[13]. Other strategies are designed to specifically block pathways involved in the acquisition of a tolerant/persister phenotype[9,14–17]. However, among the different combinations that have been clinically tested in treatment of naïve patients[4], so far, the only one that has shown a significant increase in median overall survival (OS) is with chemotherapy. Indeed, while previous studies in unselected patients failed to demonstrate such a benefit, three recent independent phase III trials in *EGFR*-mutant NSCLC patients established a clear superiority for the association of gefitinib or osimertinib with carboplatin and pemetrexed, compared to EGFR-TKI alone, albeit at the expense of a higher toxicity[18–20]. By definition, chemotherapy targets proliferating cells, so it is conceivable that, while the majority of cancer cells are inhibited by gefitinib, carboplatin and pemetrexed exert a stronger effect on the subpopulations of cells whose growth is not, or only poorly, affected by the EGFR-TKI. This combination could then function, at least in part, by preventing the emergence of gefitinib-resistant cells. These observations suggest that: (i) tumor capacity to evolve and adapt to therapy can be inhibited by neutralizing the selective advantage of resistant over sensitive cells; (ii) the capacity of resistant cells to grow in the presence of EGFR-TKIs could represent, by itself, some kind of collateral vulnerability that can be pharmacologically targeted; (iii) since it's not aimed at a specific mechanism of resistance, an approach directed against this type of vulnerability could be more broadly effective in prolonging the response to EGFR-TKIs.

Screens for new drug combinations are generally designed to find compounds that act synergistically. This type of approach is not suited to identify molecules capable of inhibiting the selective advantage of resistant cells in the presence of EGFR-TKIs. We previously devised a DNA barcoding strategy to generate and track small pools of NSCLC-resistant cells within a mass population of sensitive cells. We showed that EGFR-TKIs induce a rapid enrichment of the barcoded cells, and this effect can be blocked through specific inhibition of the mechanism of resistance[21].

Here, we show that, in combination with gefitinib, pemetrexed does not further inhibit the growth of EGFR-TKI sensitive cells, but it prevents the emergence of resistant cells. To identify other drugs capable of exerting a similar effect with potentially lower toxicity, we performed an unbiased functional screen using a NSCLC cell model containing three distinct mechanisms of resistance to EGFR-TKIs. We show that the clinically approved multikinase inhibitor sorafenib can prevent the emergence of cells displaying various types of resistance to EGFR-TKIs, including secondary *EGFR* mutations, aberrant activation of different downstream components of EGFR signaling, amplification/overexpression of other receptor tyrosine kinase, and epithelial to mesenchymal transition (EMT). Of note, sorafenib also exerts an inhibitory effect on tolerant/persister cells. Consistent with a specific vulnerability of EGFR-addicted cancer cells to sorafenib, gene expression analysis of tumor samples from various independent clinical trials indicates that patient response to this drug is significantly correlated with a high EGFR transcriptional score. Mechanistically, we demonstrate that the effects of sorafenib in NSCLC cells are independent of the MAPK pathway and rely instead on early inhibition of MKNK (also known as MNK) activity and STAT3 phosphorylation, and later downregulation of MCL1 and EGFR. Finally, using different xenograft and allograft models of acquired resistance to osimertinib, we show that the osimertinib-sorafenib combination can promote the recruitment of inflammatory cells and substantially prolong the effects of EGFR-TKI

treatment in vivo, even in extremely aggressive and rapidly progressing tumors.

## Results

### Co-treatment with sorafenib specifically inhibits the emergence of NSCLC cells resistant to EGFR-TKIs

Co-treatment with chemotherapy was recently shown to prolong NSCLC response to gefitinib[18,19]. We speculated that, in these patients, gefitinib-sensitive cancer cells are mostly inhibited by the EGFR-TKI, while chemotherapy acts on the subpopulations of resistant cells capable of growing in the presence of gefitinib. To test this hypothesis, we investigated the effects of pemetrexed and gefitinib, alone or in combination, on the viability of human NSCLC PC9 cells. While only pemetrexed inhibited the growth of gefitinib resistant cells (Supplementary Fig. 1A), the two drugs didn't exert any additive/synergistic effects in parental cells (Fig. 1A), supporting the notion that, in the presence of the combination, chemotherapy does not further affect the growth of EGFR-TKI sensitive cells, but it inhibits instead the growth of resistant cells that don't respond to EGFR-TKIs. We devised a strategy, named CRISPR-barcoding, to generate and trace resistant clones within a population of EGFR-TKI sensitive cells[21]. As shown in Fig. 1B, gefitinib treatment provoked an increase in the fraction of cells containing the resistance mutation EGFR-T790M, and this effect was blocked by co-treatment with pemetrexed. Together, these data strongly suggest that the clinical benefit of the EGFR-TKI/chemotherapy combination depends, at least in part, on its capacity to prevent or delay the emergence of EGFR-TKI-resistant cells. This association significantly prolonged the response of *EGFR*-mutant NSCLCs, but it also increased the rate of adverse effects[18,19]. We performed a small-molecule screen to identify other drugs capable of inhibiting the emergence of cells containing multiple mechanisms of resistance to EGFR-TKI, while potentially displaying lower toxicity. We used a CRISPR-barcoding cell model in which three different small pools of cells bearing a distinct resistance mutation are intermingled within a mass population of sensitive cells. As expected, the third-generation EGFR-TKI WZ42002 and the ALK-TKI TAE684 specifically blocked the enrichment of the EGFR-T790M and EML4-ALK subpopulations, respectively, but they had no effect on the other resistant cells. Intriguingly, we found that sorafenib could prevent the emergence of all three mechanisms of resistance at in vitro concentrations that were clinically relevant[22,23] (Supplementary Data 1 and Supplementary Fig. 1B, C).

We confirmed the results of our screen and showed that sorafenib markedly inhibited the gefitinib-induced increase in the proportion of EGFR-T790M mutant cells in different EGFR-addicted NSCLC lines (Fig. 1C). Third-generation irreversible EGFR-TKIs, such as osimertinib, have been developed to counter resistance induced by the T790M gatekeeper mutation[24,25]. One of the most common mechanisms of resistance to osimertinib in patients is represented by another secondary/tertiary mutation of EGFR, involving the binding site of the drug to the receptor, cysteine 797[4,26]. We modeled EGFR-C797S-mediated resistance to osimertinib in PC9 and H1975 cells, as well as in the NSCLC patient-derived cell (PDC) line YU-1150. In each case, co-treatment with sorafenib dramatically reduced the enrichment of EGFR-C797S subpopulations induced by osimertinib, regardless of whether a concurrent T790M mutation was present (Fig. 1D, E and Supplementary Fig. 1D).

Sorafenib is a multikinase inhibitor targeting RAF, the vascular endothelial growth factor receptors (VEGFR), platelet-derived growth factor receptors (PDGFR), KIT, and RET, and it has been approved for the treatment of advanced kidney, thyroid, and liver cancers[27]. Of note, a phase III trial has previously shown that sorafenib monotherapy can improve OS of patients with EGFR-mutant, but not wild-type (wt) NSCLC[28]. Consistent with these data, a transcriptional signature to detect EGFR activation was used to predict the sensitivity to sorafenib

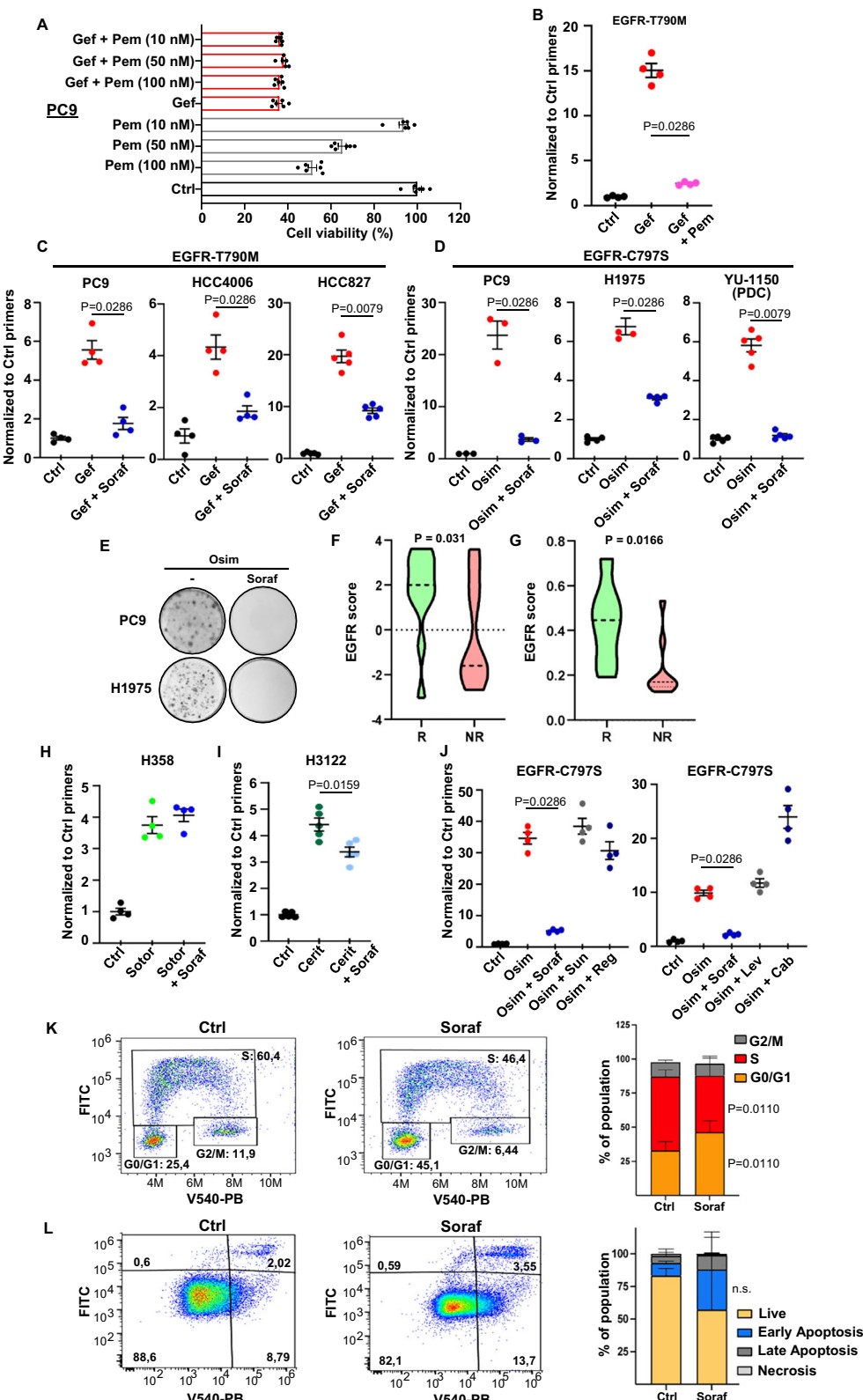

in both lung and liver cancer patients[29]. We analyzed two other data-sets containing gene expression and clinical data of cancer patients treated with sorafenib[30,31]. We found that renal cell carcinomas (Fig. 1F) and differentiated thyroid cancers (Fig. 1G) that responded to this drug displayed higher EGFR-scores. Together, these observations suggest that patients whose tumors show activation of EGFR signaling are more likely to respond to sorafenib treatment.

We next investigated whether the inhibitory effects of sorafenib on the emergence of resistance are specific of EGFR-addicted cancer cells. H358 cells derive from human NSCLC and contain the KRAS-G12C driver mutation, which makes them sensitive to a new class of clinically approved KRAS inhibitors, such as sotorasib[32]. We used CRISPR-barcoding to generate a small subpopulation of cells bearing the KRAS-G12D mutation, a mechanism of resistance to sotorasib identified in

**Fig. 1 | Sorafenib specifically inhibits the emergence of resistant subpopulations of NSCLC cells induced by EGFR-TKIs. A** Cell viability assay of PC9 cells treated for 5 d with pemetrexed (Pem) and gefitinib (1 μM; Gef). The mean ± SEM of $n = 6$ is shown (representative of three independent experiments). **B** PC9 cells containing a pool of EGFR-T790M barcoded cells were treated with gefitinib (1 μM) or pemetrexed (50 nM) for 6 d. The EGFR-T790M mean ± SEM is shown ($n = 4$; representative of three independent experiments). **C** PC9, HCC4006 or HCC827 NSCLC cells containing EGFR-T790M barcoded subpopulations were treated with gefitinib (1 μM) or sorafenib (5 μM; Soraf) for 5 d. The EGFR-T790M mean ± SEM is shown ($n = 4$; representative of three independent experiments). **D** PC9, H1975, and YU-1150 cells containing EGFR-C797S-barcoded subpopulations were treated with osimertinib (0.1 μM; Osim) alone or with sorafenib (5 μM) for 10 d. The EGFR-C797S mean ± SEM is shown ($n = 4$; representative of three independent experiments). **E** PC9 and H1975 cells described in (**E**) were treated with osimertinib (1 μM or 0.1 μM, respectively) alone or with sorafenib (5 μM) for 20 d or 15 d. The cells were then fixed and stained ($n = 3$; representative of two and three independent experiments). EGFR-scores of responders (*R*) and non-responders (*NR*) from a cohort of renal-cell carcinoma (**F**) and thyroid cancer patients (**G**) treated with sorafenib. **H** H358 cells

containing a KRAS-G12D-barcoded subpopulation were treated with sotorasib (10 nM; Sotor) alone or with sorafenib (5 μM) for 12 d. The KRAS-G12D mean ± SEM is shown ($n = 4$, representative of three independent experiments). $p = 0.4857$. **I** Ceritinib-resistant H3122 cells labeled with VIRHD lentivirus were mixed with parental cells (1:50) and treated with ceritinib (50 nM; Cerit) alone or with sorafenib (5 μM) for 7 d. VIRHD-cells mean ± SEM is shown ($n = 5$, representative of three independent experiments). **J** EGFR-C797S CRISPR-barcoded PC9 cells were treated with osimertinib (0.1 μM) alone or with sorafenib (5 μM), sunitinib (1 μM; Sun), regorafenib (2 μM; Reg), lenvatinib (1 μM; Lenv), or cabozantinib (5 μM; Cab) for 7 d (left) or 10 d (right). EGFR-C797S mean ± SEM is shown ($n = 4$–5; representative of three independent experiments). **K** PC9 cells were treated with sorafenib (5 μM) for 72 h, and cell cycle was analyzed by FACS. The mean ± SEM of four independent experiments is shown. **L** PC9 cells were treated with sorafenib (5 μM) for 72 h, and percentage of apoptotic cells was measured by annexin V staining. The mean ± SEM of three independent experiments is shown. ns not significant. Statistics are: **B**, **C**, **D**, **H**, **I**, **J** matched Mann-Whitney, two-tailed; **F**, **G** matched Student *t*-test, two-tailed; **K**, **L** matched two-way ANOVA with the Tukey correction. N refers to biological replicates. Source data are provided as a Source Data file.

patients[33]. As shown in Fig. 1H, treatment with sotorasib induced an enrichment of the proportion of KRAS-G12D cells, which was not affected by co-treatment with sorafenib. Through a similar approach, we generated a model of resistance to the ALK inhibitor ceritinib in the H3122 NSCLC cell line, harboring the *EML4-ALK* inversion. We found that co-treatment with sorafenib only mildly reduced the enrichment of resistant cells induced by ceritinib (Fig. 1I). Together, our data indicate that sorafenib can effectively prevent the emergence of resistance in cancer cells addicted to mutant *EGFR*, but not in cells addicted to other driver mutations frequently found in NSCLC, such as those involving the oncogenes *KRAS* and *ALK*.

Other clinically approved multikinase inhibitors, including sunitinib, regorafenib, lenvatinib, and cabozantinib, show partially overlapping pharmacological profiles with sorafenib. As shown in Fig. 1J, none of these other compounds could inhibit the enrichment of EGFR-C797S resistant cells induced by osimertinib, indicating that the effects of sorafenib are specific and cannot be mimicked by other multikinase inhibitors.

To investigate how sorafenib inhibits the growth of NSCLC cells, we analyzed cell cycle and apoptosis by flow cytometry. We found that, in PC9 cells, sorafenib inhibited proliferation and promoted G1 growth arrest (Fig. 1K), without affecting cell survival (Fig. 1L). This drug also repressed proliferation and promoted apoptosis in another NSCLC cell line tested (Supplementary Fig. 1E, F).

**Sorafenib prevents the enrichment of cells containing different, clinically relevant mechanisms of resistance to EGFR inhibition**
The results of our initial screen suggested that sorafenib can act on different types of EGFR-TKI-resistant cells. We used CRISPR-barcoding to model other resistance mechanisms found in NSCLC patients[4,34,35]. As shown in Fig. 2A, co-treatment with sorafenib inhibited the osimertinib-induced enrichment of PC9 subpopulations containing other types of resistance mutations involving receptor tyrosine kinases (RTKs), including EGFR-G724S or an insertion in exon 20 of *ERBB2* (ERBB2-ex20ins). We then investigated the effects of this drug on cells containing activating mutations of different downstream components of the EGFR signaling. While unable to reshape the oncogenic program triggered by EGFR in these cells, this type of mutation can confer a selective advantage in the presence of EGFR-TKIs. We found that sorafenib prevented the emergence of cells harboring KRAS-G12D, BRAF-V600E, or PIK3CA-E545K, in both PC9 cells (Fig. 2A) and other NSCLC cell models, including PDCs (Fig. 2B and Supplementary Fig. 2A–C). Of note, co-treatment with the MEK inhibitor trametinib could inhibit cells containing only some of these mutations (Fig. 2A).

Consistent with our initial screen, sorafenib also inhibited the amplification of cells containing the chromosomal inversion leading to

the expression of the EML4-ALK fusion oncogene (Fig. 2C), suggesting that this combination could also be effective against resistant cells expressing oncogenes originating from RTK fusion, which has been described in a few patients progressing to EGFR-TKIs[4,34,35].

Amplification of other RTKs, including *MET* and *ERBB2*, is among the most common mechanisms of resistance to osimertinib[4,34,35]. We modeled these aberrations in PC9 cells using lentivirus-mediated cDNA overexpression (Supplementary Fig. 2D) or upregulation of the endogenous gene through a dCas9 activator system[36] (Supplementary Fig. 2E). As shown in Fig. 2D, co-treatment with sorafenib strongly inhibited the emergence of cells overexpressing either ERBB2-ex20ins or MET. To confirm these results, we performed a mixing experiment using parental HCC827 and lentiviral-labeled HCC827-GR6 cells, a *MET*-amplified clone derived by long-term selection in the presence of gefitinib[37] (Supplementary Fig. 2F). Figure 2E shows that the enrichment of the HCC827-GR6 subpopulation induced by osimertinib was strongly reduced by co-treatment with sorafenib, indicating that this drug can antagonize acquired resistance mediated by *MET* amplification.

It has been reported that resistance to EGFR-TKIs can be accompanied by typical features of an epithelial to mesenchymal transition (EMT)[4,34]. Using a lentiviral inducible system (Supplementary Fig. 2G), we showed that expression of the EMT transcription factor SNAI2 (also known as SLUG) can confer resistance to osimertinib in PC9 cells, and this effect was abrogated by co-treatment with sorafenib (Fig. 2F). These data imply that this combination can also affect cells that escape EGFR-TKI treatment through phenotypic transformation.

Anti-EGFR therapies are used for other tumor types, including colorectal cancers (CRCs) with wt *RAS* and *BRAF*, which are treated with anti-EGFR monoclonal antibodies, such as cetuximab or panitumumab. Similarly to NSCLCs, these tumors eventually relapse, due to the acquisition of resistance through various mechanisms, including activation of downstream components of the EGFR pathway or mutations in the extracellular domain of the receptor that prevent its recognition by the antibodies[38]. We used CRISPR-barcoding to model these mechanisms of resistance in CRC cells sensitive to EGFR inhibition. We found that co-treatment with sorafenib blocked the enrichment of CRC cells containing mutations of either EGFR extracellular domain or KRAS induced by cetuximab (Fig. 2G, H), indicating that sorafenib could prolong the response to targeted therapy in other types of EGFR-addicted tumors.

**Early inhibition by sorafenib of MKNK activity, STAT3 phosphorylation, and MCL1 expression in NSCLC cells**
Despite having been originally developed as an inhibitor of the serine/threonine kinase RAF, in PC9 cells sorafenib was unable to down-

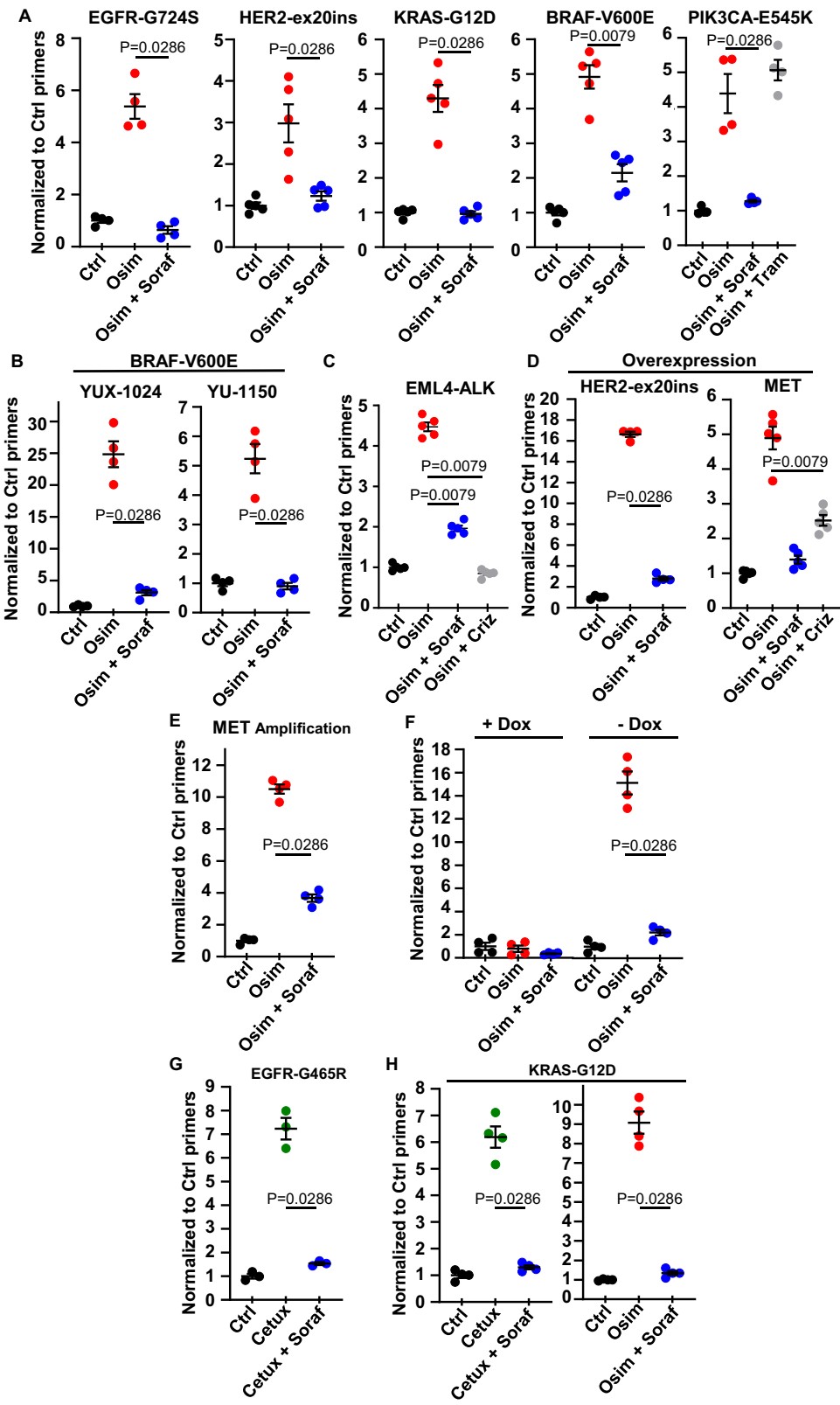

regulate phospho-ERK levels (Fig. 3A) or the expression of well-characterized MAPK-regulated genes, including *DUSP6*, *SPRY2* and *ETV5*[39], as shown by qPCR (Supplementary Fig. 3A). Instead, as previously reported in leukemia cells[40], sorafenib rapidly inhibited phosphorylation of the eukaryotic translation initiation factor 4E (eIF4E) in PC9 cells (Fig. 3A), as well as other NSCLC cells, including two different PDC lines (Fig. 3B and Supplementary Fig. 3B, C), and CRC cells

(Supplementary Fig. 3D). A component of the translation regulatory machinery, eIF4E has been implicated in different types of cancer[41–44] and its phosphorylation is mediated by MKNKs, whose activity is regulated by ERK[41]. Because sorafenib did not detectably affect MAPKs in NSCLC cells and its effects on eIF4E phosphorylation were readily seen after 30 min of treatment (Fig. 3A), we hypothesized that this multikinase inhibitor might act directly on MKNKs. A potential

**Fig. 2 | Sorafenib specifically abolishes the selective advantage of cancer cells containing different clinically relevant mechanisms of resistance to EGFR inhibition. A** PC9 cells containing subpopulations of EGFR-G724S, ERBB2-ex20ins, KRAS-G12D, BRAF-V600E or PIK3CA-E545K cells were treated as indicated with osimertinib (0,1 μM; Osim), sorafenib (5 μM; Soraf) or trametinib (10 nM; Tram) for 7 d to 20 d. The mean ± SEM of the mutant barcodes is shown (n = 4; representative of three independent experiments). **B** BRAF-V600E osimertinib-resistant YUX-1024 and YU-1150 cells were mixed with parental cells (1:100) and treated for 7 d with osimertinib (0.1 μM) alone or with sorafenib (5 μM). The BRAF-V600E mean ± SEM is shown (n = 4; representative of three independent experiments). **C** PC9 cells containing a EML4-ALK-barcoded subpopulation were treated for 10 d with osimertinib (0.1 μM), sorafenib (5 μM), or crizotinib (500 nM; Criz). The EML4-ALK mean ± SEM is shown (n = 5; representative of three independent experiments). **D** PC9 cells overexpressing ERBB2-ex20ins or MET were mixed with parental cells (1:100) and treated with osimertinib (0.1 μM) alone or with sorafenib (5 μM) for 10 d or 15 d. The mean fraction ± SEM of ERBB2-ex20ins- or MET-overexpressing cells is

shown (n = 4; representative of three independent experiments). **E** Lentiviral-labeled, osimertinib-resistant HCC827-GR6 cells were mixed with parental HCC827 (1:100) and treated for 6 d with osimertinib (0.1 μM) alone or with sorafenib (5 μM). HCC827-GR6 mean fraction ± SEM is shown (n = 4; representative of three independent experiments). **F** PC9 cells transduced with empty or inducible-SNAI2 vectors were pre-treated with or without doxycycline (1 μg/ml; Dox) for 7 d, mixed with parental PC9 (1:100) and treated for 7 d with osimertinib (0.1 μM) alone or with sorafenib (5 μM). The mean fraction ± SEM of vector-labeled cells is shown (n = 4; representative of three independent experiments). LIM1215 cells containing EGFR-G465R (**G**) or KRAS-G12D (**H**) CRISPR-barcodes were treated for 6 d with cetuximab (20 μg; Cetux) or osimertinib (1 μM), alone or with sorafenib (5 μM). The barcode mean fraction ± SEM is shown (n = 3 for EGFR-G465R; n = 4 for KRAS-G12D; representative of three independent experiments). Statistics calculated by Mann–Whitney test, two-tailed. "N" refers to biological replicates. Source data are provided as a Source Data file.

inhibition of MKNK catalytic activity by sorafenib has been suggested by high-throughput in vitro kinase screens[45], although, to our knowledge, this has not been demonstrated in living cells.

We initially confirmed that sorafenib can affect MKNK1/2 activity in in vitro assays, in which the myelin basic protein or a short peptide derived from the human cAMP Response Element Binding protein was used as a substrate (Supplementary Fig. 3E, F). To investigate the potential direct effects of sorafenib on MKNK in PC9 cells, we generated a lentiviral inducible system to express a constitutively active form of MKNK2 (CA-MKNK2). Figure 3C shows that sorafenib, but not the MEK inhibitor trametinib, dose-dependently inhibited the phosphorylation of eIF4E induced by CA-MKNK2. In accordance with these results, in vitro phosphorylation of immunoprecipitated FLAG-tagged eIF4E by recombinant MKNK2 was inhibited by sorafenib, but not by trametinib (Fig. 3D). Together, our data demonstrate that sorafenib can block eIF4E phosphorylation by directly inhibiting MKNK catalytic activity. Of note, MKNK inhibition by sorafenib didn't affect cap binding of eIF4E (Supplementary Fig. 3G), suggesting that eIF4E phosphorylation doesn't exert a global effect on protein translation, consistent with previous studies[41].

After a few hours of treatment, sorafenib also inhibited the phosphorylation of STAT3 on tyrosine 705 (Fig. 3E and Supplementary Fig. 4A), as previously described in other types of cancer[46]. Of note, while cabozantinib reduced eIF4E phosphorylation, probably through MAPK inhibition, and regorafenib moderately affected phospho-STAT3, only sorafenib was able to inhibit both eIF4E and STAT3 phosphorylation (Supplementary Fig. 4B, C). Sorafenib also induced a more delayed down-regulation of MCL1 in different NSCLC cells, including two PDC lines (Fig. 3F, G and Supplementary Fig. 4D). To investigate the mechanism involved in sorafenib-induced inhibition of MCL1, we tested the effects of this drug on MCL1 transcription by qPCR. As shown in Supplementary Fig. 4E, MCL1 mRNA levels were not reduced in the presence of sorafenib, ruling out a transcriptional effect. To assess whether sorafenib could affect MCL1 translation, we co-treated the cells with the protein synthesis inhibitor cycloheximide. Figure 3H shows that sorafenib could still downregulate MCL1 even when the synthesis of new proteins is blocked, implying that this drug affects MCL1 at a post-translational level. We then investigated whether sorafenib could enhance the degradation of MCL1 protein by co-treating the cells with the lysosome inhibitor chloroquine or the proteasome inhibitor MG132. While chloroquine had no effects, MG132 abolished the capacity of sorafenib to inhibit MCL1 (Fig. 3I). Together, these studies indicate that sorafenib inhibits MCL1 by promoting its proteasomal degradation.

To investigate whether the inhibition of MKNK, STAT3, and MCL1 could play a role in the effects of sorafenib in NSCLC cells, we tested specific inhibitors of these three factors (Supplementary Fig. 4F, G). While the emergence of resistant cells could not be blocked by eFT-508, napabucasin, or S63845 alone, we found that the

combination of all three compounds mimicked the effects of sorafenib (Fig. 3J). To confirm these findings, we engineered osimertinib-resistant PC9 cells containing the EGFR-C797S mutation to express under the control of doxycycline shRNAs targeting STAT3 and MCL1, as well as a dominant-negative MKNK1 (DN-MKNK1)[43] fused to GFP (Supplementary Fig. 4H). We then mixed a small pool of these cells with parental PC9. As shown in Fig. 4K, osimertinib induced an increase in the proportion of resistant cells, and this effect was blocked by co-treatment with sorafenib. Of note, the enrichment of resistant cells was also inhibited by doxycycline-induced concurrent downregulation of STAT3 and MCL1 expression and MKNK activity. Together, these results indicate that sorafenib prevents the emergence of EGFR-TKI resistant subpopulations of NSCLC cells through a mechanism involving, at least in part, its combined inhibitory effects on MKNK, STAT3, and MCL1.

## Late down-regulation of EGFR by sorafenib in NSCLC cells

Time-course experiments in PC9 cells revealed that sorafenib induced a marked down-regulation of EGFR expression at the protein level by 2–3 days of treatment, with concomitant MAPK inhibition (Fig. 4A). EGFR down-regulation was also observed in other NSCLC cells, including PDCs, as well as in HCA46 CRC cells (Fig. 4B and Supplementary Fig. 5A), while the other multikinase inhibitors sunitinib and regorafenib did not inhibit the receptor or had much weaker effects compared to sorafenib (Supplementary Fig. 5B). To test whether sorafenib could exert similar effects in vivo, we treated with this drug SCID mice bearing PC9 tumors. We found that tumors from sorafenib-treated mice showed a decreased expression of both EGFR and MCL1 (Fig. 4C), consistent with our in vitro studies.

We then investigated the mechanism responsible for sorafenib-induced inhibition of EGFR. As shown in Supplementary Fig. 5C, EGFR mRNA levels were not affected by this drug, implying a post-transcriptional effect. To investigate whether sorafenib inhibited EGFR protein translation or stability, we generated a lentiviral vector in which the PGK promoter drives expression of FLAG-tagged EGFR and green fluorescent protein (GFP), separated by the T2A self-cleaving peptide. Sorafenib strongly reduced the expression of the exogenous receptor, whereas it had no effect on GFP levels (Fig. 4D), indicating that the multikinase inhibitor does not affect EGFR translation, but instead promotes EGFR degradation.

It has been reported that, in NSCLC cells, mutant EGFR, but not wt, can be specifically degraded by a macropinocytosis-dependent lysosomal pathway[47]. Consistent with this mechanism, the inhibitory effects of sorafenib were more pronounced on the Ex19Del mutant compared to the wt receptor (Fig. 4E). We also found that suppression of lysosomal degradation by chloroquine blocked the down-regulation of EGFR induced by sorafenib (Fig. 4F). Of note, co-treatment with chloroquine also partially rescued the effects of this multikinase inhibitor on the emergence of EGFR-TKI-resistant cells (Fig. 4G), implying

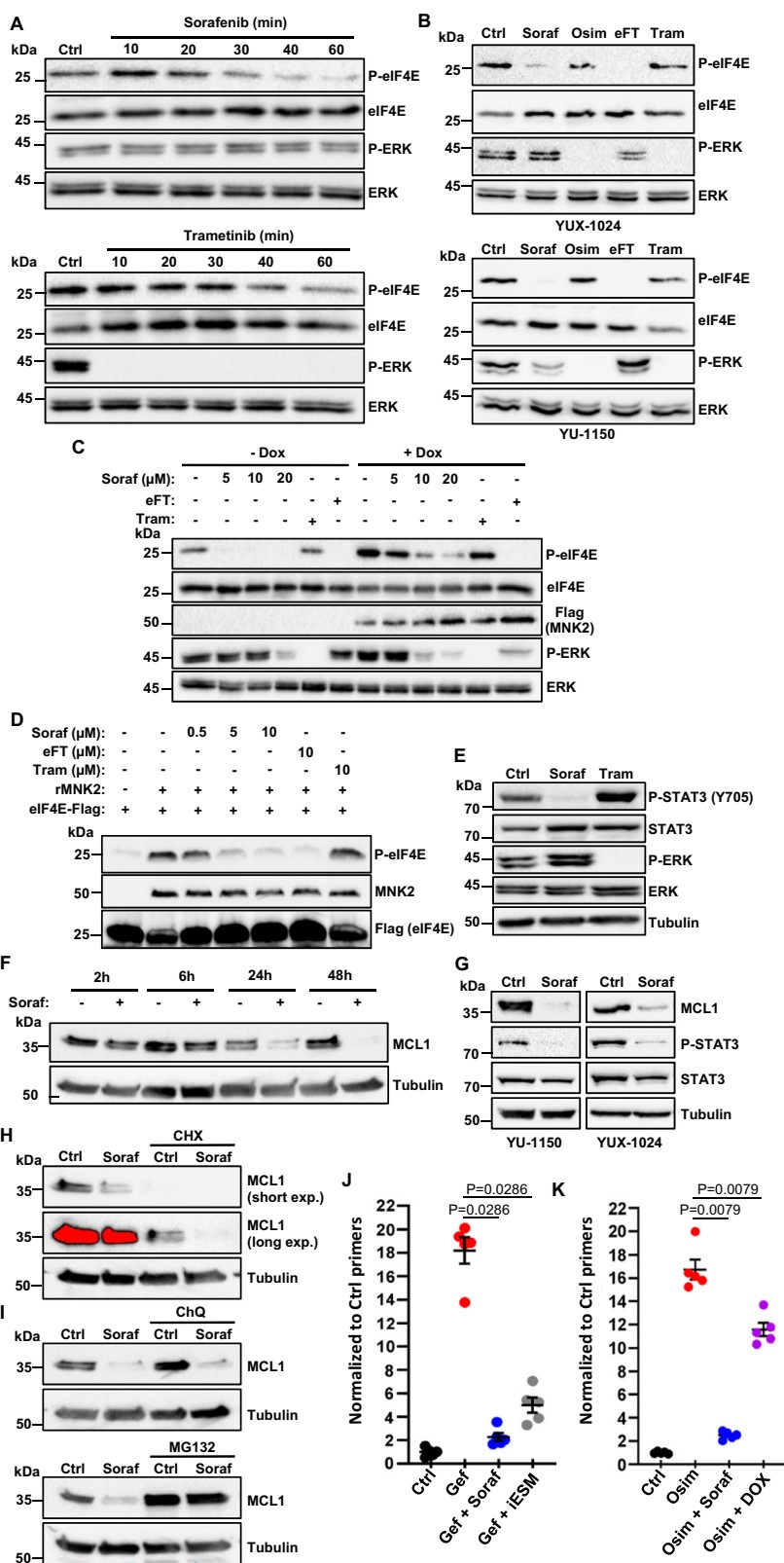

that its ability to induce EGFR down-regulation contributes to the mechanism of action of sorafenib in NSCLC cells.

**Co-treatment with sorafenib inhibits the selective advantage of EGFR-TKI-resistant cells**

Through our barcoding models, we showed that co-treatment with sorafenib can inhibit the amplification of EGFR-TKI resistant cells.

These experiments were based on mixed populations in which small pools of EGFR-TKI resistant cells were intermingled within a majority of EGFR-TKI sensitive cells. To separately compare the effects of sorafenib and EGFR-TKIs on either sensitive or resistant cells, we generated a homogeneous population of osimertinib-resistant PC9 cells containing the EGFR-C797S mutation (OR-PC9). As expected, osimertinib inhibited the growth of parental, but not OR-PC9 cells,

**Fig. 3 | Sorafenib prevents NSCLC cell resistance to EGFR-TKIs independently of MAPKs by inhibiting MKNK activity, STAT3 phosphorylation, and MCL1 expression. A** PC9 cells were treated with sorafenib (5 µM; Soraf) or trametinib (50 nM; Tram), followed by immunoblot (representative of three independent experiments). **B** YUX-1024 and YU-1150 PDCs were treated for 2 h with sorafenib (5 µM), osimertinib (0.1 µM; Osim), eFT-508 (1 µM; eFT), or trametinib (50 nM), followed by immunoblot (representative of three independent experiments). **C** PC9 cells containing inducible constitutively active MKNK2 (CA-MKNK2; Flag-tagged) were pre-treated with or without doxycycline (0.1 µg/ml; Dox) for 12 h, treated for 1 h with sorafenib, eFT-508 (20 µM) or trametinib (1 µM), followed by immunoblot (representative of four independent experiments). **D** Flag-eIF4E was incubated with recombinant active MKNK2 (rMKNK2) and the indicated inhibitors, followed by immunoblot (representative of three independent experiments). **E** PC9 cells were treated with sorafenib (5 µM) or trametinib (50 nM) for 6 h, followed by immunoblot (representative of three independent experiments). **F** PC9 cells were treated with or without sorafenib (5 µM), followed by immunoblot (representative of four independent experiments). **G** YU-1150 and YUX-1024 PDCs were treated with

sorafenib (5 µM) for 3 d, followed by immunoblot (representative of three independent experiments). **H** PC9 cells were treated with sorafenib (5 µM) in the presence or the absence of cycloheximide (20 µg/mL; CHX) for 1 d, followed by immunoblot (representative of three independent experiments). **I** PC9 cells were treated with sorafenib (5 µM), chloroquine (50 µM; ChQ) or MG132 (5 µM) for 1 d as indicated, followed by immunoblot (representative of three independent experiments). **J** EGFR-T790M CRISPR-barcoded PC9 cells were treated for 5 d as indicated with gefitinib (1 µM), sorafenib (5 µM), or a combination (iESM) of napabucasin (0.5 µM), S63845 (0.1 µM), and eFT-508 (1 µM). EGFR-T790M mean ± SEM is shown (n = 5; representative of three independent experiments). **K** Osimertinib-resistant PC9 cells (EGFR-C797S) containing shMCL1, shSTAT3, and DN-MKNK1 inducible vectors were mixed with parental PC9 (1:100) and treated for 6 d as indicated with osimertinib (100 nM), sorafenib (5 µM), or doxycycline (2 µg/ml). EGFR-C797S mean ± SEM is shown (n = 5, representative of three independent experiments). **J, K** p values were calculated by Mann–Whitney two-tailed test and "n" refers to biological replicates. Source data are provided as a Source Data file.

whereas sorafenib similarly affected both sensitive and resistant cells (Fig. 5A). We then tested the effects of the osimertinib-sorafenib combination in each cell line. As illustrated in Fig. 5A, the combination affected the growth of OR-PC9 cells to the same extent than sorafenib alone. In parental PC9, the osimertinib-sorafenib association did not further inhibit cell growth compared to the two compounds alone. A similar lack of additive/synergistic effects was observed for the gefitinib-sorafenib combination (Supplementary Fig. 6A), and it probably explains why this multikinase inhibitor was not considered a hit in previous drug screens aimed at identifying new combinations to prevent resistance in NSCLC cells[48]. Intriguingly, we found that cabozantinib exerted an additive/synergistic effect with osimertinib on sensitive cells, while it only poorly affected resistant cells (Fig. 5A). In a mixed population containing both EGFR-TKI sensitive and resistant cells, it is then conceivable to speculate that the cabozantinib-osimertinib combination would exert a stronger effect on sensitive cells, thus further promoting the emergence of resistant cells. This is consistent with the results obtained in our barcoding experiments, in which, contrary to sorafenib, cabozantinib enhanced, instead of inhibiting, the enrichment of EGFR-C797S cells induced by osimertinib (Fig. 1J).

We then compared the effects of sorafenib in NSCLC cells addicted to other oncogenes. Sorafenib alone similarly inhibited the growth of PC9 (EGFR addicted), H358 (KRAS addicted), and H3122 (ALK addicted) cells. However, contrary to what was observed in PC9 cells, sorafenib displayed an additive effect with sotorasib and ceritinib in H358 and H3122 cells, respectively (Fig. 5B). These results imply that, when combined with sotorasib or ceritinib, sorafenib would inhibit the growth of both sensitive and resistant cells, without changing the relative proportion between these two cell populations, as we showed in our previous barcoding experiments in H358 and H3122 cells (Fig. 1H, I). Conversely, in a population of EGFR-addicted cells, adding sorafenib to osimertinib would inhibit the growth of resistant cells, without further affecting sensitive cells. This would then result in a reduction of the selective advantage of resistant cells in the presence of osimertinib. This model is consistent with the results of our barcoding studies in EGFR-addicted cancer cells.

To gain insights into the mechanism responsible for the lack of additive effects of sorafenib and osimertinib in NSCLC cells, we investigated how these compounds, alone or in combination, affect MCL1, eIF4E, and EGFR in osimertinib-sensitive and resistant cells. While both osimertinib and sorafenib inhibited eIF4E phosphorylation, albeit through a different mechanism, as discussed above, sorafenib, but not osimertinib, downregulated MCL1 expression in sensitive and resistant cells. Of note, in sensitive cells, MCL1 inhibition in the presence of the sorafenib-osimertinib combination was slightly less pronounced as compared to sorafenib alone (Fig. 5C). Sorafenib also

inhibited EGFR expression to the same extent in sensitive and resistant cells. However, as shown in Fig. 5D, EGFR downregulation by sorafenib was abolished by co-treatment with osimertinib in the sensitive, but not the resistant cells. Similar results were obtained with the gefitinib-sorafenib combination (Supplementary Fig. 6B). Together, these data indicate that, in sensitive cells treated with the osimertinib-sorafenib combination, the status of sorafenib downstream effectors is similar to what can be observed in the presence of osimertinib alone. This is consistent with the lack of additive effects by the two drugs and suggests that the growth inhibition of these cells is mostly due to the effects of osimertinib.

To confirm this model, we investigated the effects of these drugs, alone or in combination, on the gene expression profile of PC9 cells (Supplementary Data 2). As shown by gene set enrichment analysis (GSEA), the profile of osimertinib-treated cells correlated very well with a previously published EGFR-TKI signature[49]. A similar result was obtained for cells treated with the combination, whereas the enrichment score (ES) was negative in the presence of sorafenib (Fig. 5E and Supplementary Fig. 7A), indicating that, when sensitive cells are exposed to both drugs, the effects on EGFR-dependent gene expression are predominantly mediated by osimertinib. Consistent with these observations, when we generated our own signature for the osimertinib/sorafenib combination (Supplementary Data 3), we found higher ES for cells treated with osimertinib, as compared to sorafenib (Fig. 5F and Supplementary Fig. 7B). These results were also confirmed by cluster analysis (Supplementary Fig. 7C) and suggest that, in a mixed cell population treated with both drugs, EGFR-TKI sensitive cells respond primarily to osimertinib, whereas sorafenib acts on resistant cells, which, by definition, cannot be affected by osimertinib (Supplementary Fig. 8).

## The osimertinib/sorafenib combination delays NSCLC resistance in vivo

We next sought to investigate the effects of the osimertinib/sorafenib combination in vivo. Conventional strategies are generally ill-suited to monitor acquired resistance to this EGFR-TKI in NSCLC mouse models, since the initial response is particularly long, typically lasting for several months[24]. We thus performed experiments using mass populations constituted of a large majority of osimertinib-sensitive cells and small pools of resistant cells. Using iDISCO 3D imaging[50], we showed that EGFR-TKI resistant PC9 cells containing EGFR-C797S, KRAS-G12D, or PIK3CA-E545K mutations were enriched in xenografts from mice treated with osimertinib, and this effect was inhibited by co-treatment with sorafenib (Fig. 6A, Supplementary Fig. 9A and Supplementary Movie 1). To better estimate the tumor growth delay induced by this drug combination, SCID mice inoculated with EGFR-C797S CRISPR-barcoded PC9 cells were treated with

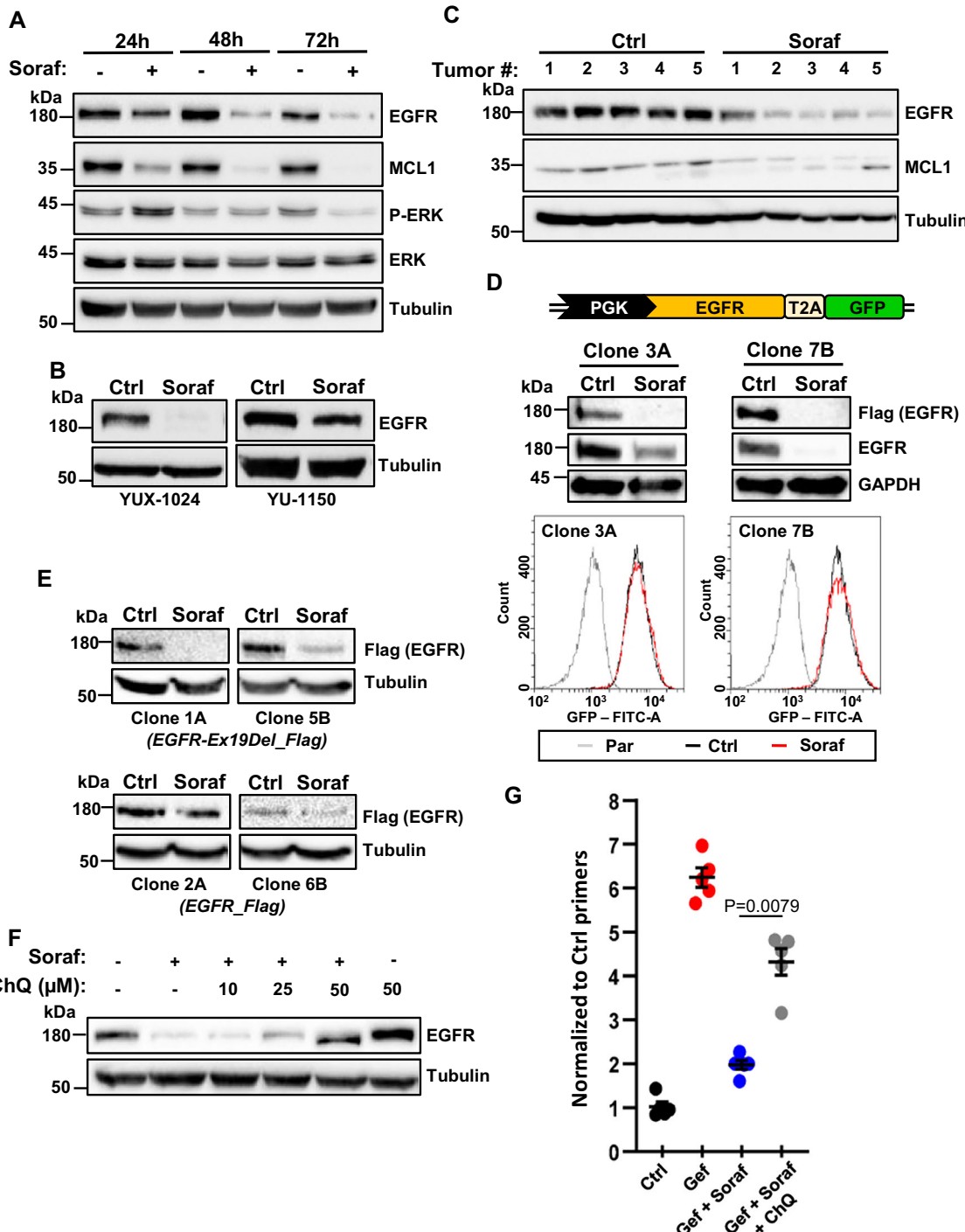

**Fig. 4 | Lysosomal degradation of EGFR participates in the effects of sorafenib in preventing the emergence of EGFR-TKI-resistant NSCLC cells. A** Time-course effects of sorafenib (5 μM; Soraf) in PC9 cells. Immunoblot was performed using the indicated antibodies (representative of four independent experiments). **B** YUX-1024 and YU-1150 PDCs were treated with sorafenib (5 μM) for 5 d, followed by immunoblot (representative of three independent experiments). **C** PC9 cells were injected in the flanks of female SCID mice and, once the tumors reached a mean volume of about 200 mm³, the mice were randomized and treated with control vehicle or sorafenib (60 mg/kg) for 4 d. The mice were then sacrificed, and the tumors dissected. Immunoblot was performed using the indicated antibodies (n = 5 mice). **D** PC9 cells were transduced with a lentiviral vector containing Flag-tagged EGFR-Ex19Del and the green fluorescent protein (GFP), separated by the T2A self-cleaving peptide and under the control of the PGK promoter. Two different clones were isolated and treated for 3 d in the presence or the absence of sorafenib (5 μM).

The expression of EGFR and GFP was measured by immunoblot (upper panel) and FACS (lower panel), respectively. Representative blots and FACS analysis from three independent experiments. **E** PC9 cells were transduced with lentiviral vectors containing either Ex19Del-mutant or wt Flag-tagged EGFR, and two clones per condition were isolated. The clones were treated with or without sorafenib (5 μM) for 3 d, followed by immunoblot (representative of four independent experiments). **F** PC9 cells were treated with sorafenib (5 μM) alone or with different concentrations of chloroquine (ChQ) for 3 d, followed by immunoblot (representative of three independent experiments). **G** PC9 cells containing the EGFR-T790M CRISPR-barcode were treated with gefitinib (1 μM; Gef) alone or in combination with sorafenib (5 μM), with or without chloroquine (20 μM) for 4 d. The EGFR-T790M mean fraction ± SEM is shown (n = 5 biological replicates; representative of three independent experiments). Statistics are Mann–Whitney test, two-tailed. Source data are provided as a Source Data file.

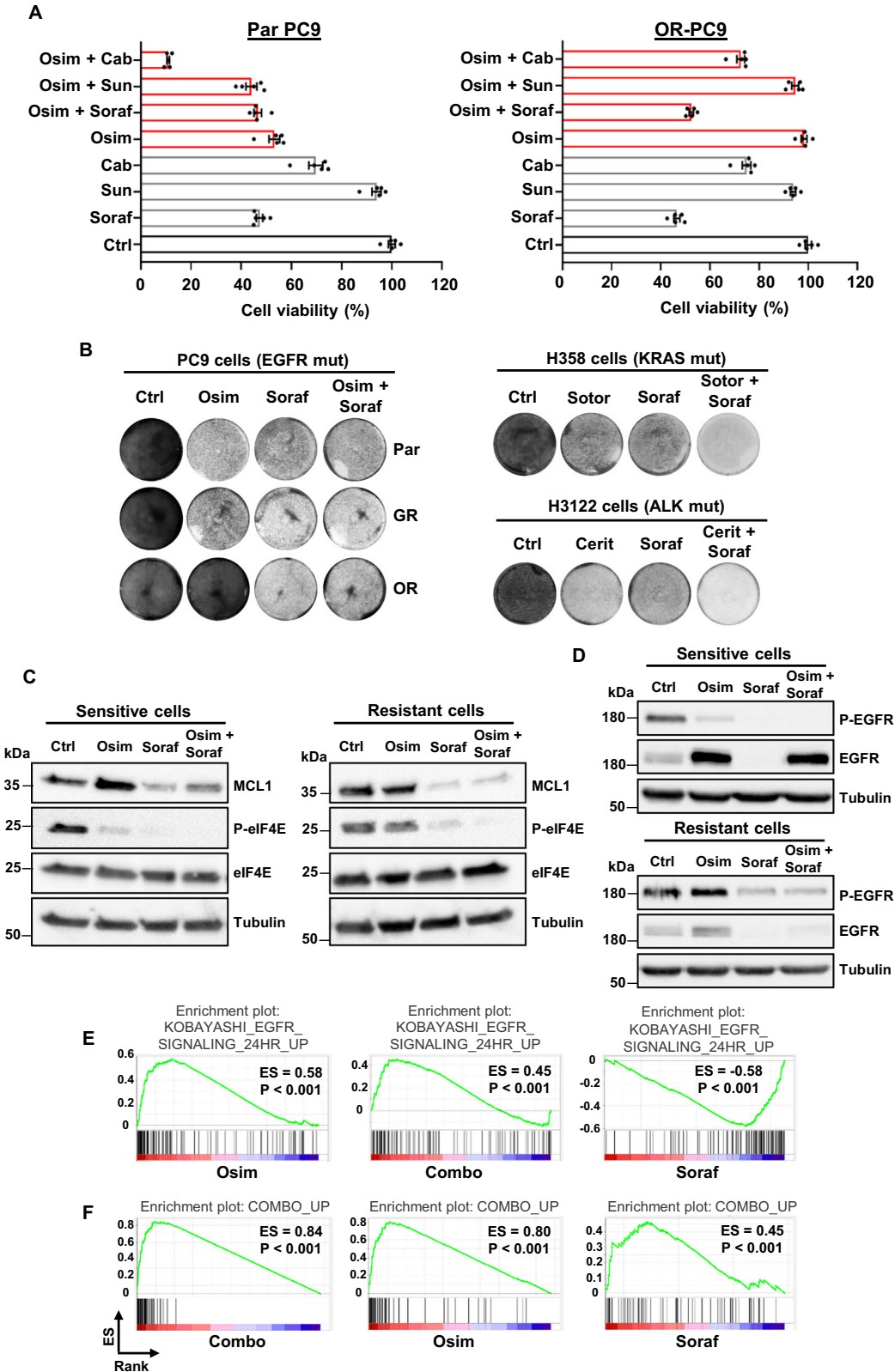

osimertinib and/or sorafenib. As illustrated in Fig. 6B and Supplementary Fig. 9B, the growth of the tumors was slowed by sorafenib and, more markedly, by osimertinib, while it was completely blocked by the combination.

The PDC line YUX-1024 forms tumors that show some similarities with patient-derived xenografts obtained from the same patient (Supplementary Fig. 9C). We mixed a small fraction of barcoded BRAF-

V600E-mutant cells in a mass population of YUX-1024 cells, which was then inoculated in SCID mice. Osimertinib delayed tumor growth for about a month, while the osimertinib-sorafenib combination was considerably more effective, slowing progression for 3 additional months (Fig. 6C–E, Supplementary Fig. 9D and Supplementary Movie 2). Of note, this regimen could be maintained for more than 5 months without any sign of toxicity.

**Fig. 5 | Effects of sorafenib and osimertinib in osimertinib-sensitive and resistant cells. A** Parental (Par) or osimertinib-resistant (OR) PC9 cells were treated for 5 d as indicated with osimertinib (0,1 μM; Osim), sorafenib (5 μM; Soraf), sunitinib (1 μM; Sun) or cabozantinib (5 μM; Cab). The cell viability mean ± SEM is shown (n = 5 biological replicates; representative of four independent experiments). **B** Left panel: colony forming assay representative images of parental, gefitinib-resistant/osimertinib-sensitive (GR) or osimertinib-resistant PC9 cells treated for 5 d with osimertinib (1 μM) or sorafenib (5 μM), alone or in combination. Right panel: colony forming assay of H358 cells treated for 8 d with sotorasib (10 nM; Sotor) or sorafenib (5 μM), alone or in combination, or H3122 cells treated for 7 d with ceritinib (50 nM; Cerit) or sorafenib, alone or in combination (n = 3 biological replicates;

representative of three independent experiments). Osimertinib-sensitive and resistant PC9 cells were treated for 1 d (**C**) or 3 d (**D**) with osimertinib (1 μM), sorafenib (5 μM), or the combination, followed by immunoblot using the indicated antibodies (representative of three independent experiments). **E** Gene set enrichment analysis (GSEA) of genes up-regulated by EGFR-TKIs in NSCLC cells (KOBAYASHI_EGFR_SIGNALING_24HR_UP signature), performed on gene array data obtained from PC9 cells treated with osimertinib (1 μM) or sorafenib (5 μM), alone or in combination (Combo) for 2 d. Enrichment scores (ES) and p values are reported (two-sided permutation test). **F** The data described in (**E**) were analyzed using our osimertinib-sorafenib combination signature (COMBO_UP). Source data are provided as a Source Data file.

## Co-treatment with sorafenib prolongs the response to osimertinib in highly aggressive models of resistance in both immunodeficient and immunocompetent mice

The duration of osimertinib response as a single agent is relatively long[51], which makes it more difficult to set up clinical trials to test its effects in combination with other drugs. To investigate whether co-treatment with sorafenib can inhibit the growth of tumors rapidly progressing on osimertinib, we developed a more aggressive model of acquired resistance, which mimics the late phases of osimertinib responsiveness and the beginning of tumor relapse. We generated a PC9 population containing four distinct pools of resistant cells, bearing either EGFR-C797S, PIK3CA-E545K or KRAS-G12D, or overexpressing ERBB2-ex20ins (Supplementary Fig. 10A). Upon injection in SCID mice, these cells formed tumors whose growth was inhibited by osimertinib by only 2 weeks, with the rapid amplification of the different subpopulations of barcoded resistant cells (Supplementary Fig. 10B). As shown in Fig. 7A, B, co-treatment with sorafenib further delayed tumor growth by more than a month, suggesting that this combination could also be beneficial in patients showing early signs of progression to osimertinib.

EGFR-mutant NSCLCs have been reported to respond poorly to anti-PD1/PDL1 inhibitors[52]. However, it has been shown that EGFR signaling can promote a noninflamed tumor microenvironment, and inhibition of this receptor can stimulate the infiltration of inflammatory immune cells[53–55]. These observations suggest that acquisition of EGFR-TKI resistance could be influenced by the presence of a functional immune system. We first tested whether treatment of NSCLC cells with osimertinib or sorafenib could affect the expression of immune-related genes. As shown in Fig. 7C, GSEA revealed up-regulation of an inflammatory response signature in PC9 cells treated with both inhibitors, either alone or in combination, suggesting that these drugs might enhance tumor recognition by the immune system.

Transgenic models are poorly suited to investigate acquired resistance, and the few available mouse NSCLC lines don't contain mutant EGFR. To investigate whether sorafenib might prolong the response to osimertinib in immunocompetent mice, we generated a new syngeneic model of oncogenic addiction to mutant-EGFR by transforming BALB-3T3 cells[56] using a lentiviral vector containing mouse Egfr-L860R (corresponding to human EGFR-L858R). Compared to cells expressing GFP or a dominant-negative Trp53 (DN-Trp53), BALB-3T3-Egfr-L860R cells are highly sensitive to osimertinib (Supplementary Fig. 11A). Consistent with these data, these cells can be made unresponsive to EGFR-TKIs upon insertion of a resistance mutation by CRISPR-barcoding (Supplementary Fig. 11B). As shown in Supplementary Fig. 11C, BALB-3T3-Egfr-L860R cells formed rapidly growing tumors in BALB/c mice. To obtain a homogeneous population of tumorigenic cells, we dissected the tumors and isolated several clones sensitive to osimertinib from the disaggregated cells (Supplementary Fig. 11D). We then took one of these clones (BALB-3T3-Egfr-L860R-M5, hereafter BEM-5 cells) and generated by CRISPR-barcoding a subpopulation of osimertinib resistant cells containing the mouse Egfr-C799S mutation (corresponding to EGFR-C797S in the human

receptor). Upon injection in immunocompetent mice, these cells formed aggressive tumors that showed short-term response to osimertinib. To investigate the effects of treatment on the recruitment of immune cells, once the tumors reached a volume of about 50 mm³ the mice were treated for 10 days in the presence or the absence of osimertinib and sorafenib, alone or in combination, followed by immunohistochemistry analysis. As shown in Fig. 7D and Supplementary Fig. 11E, treatment with the inhibitors increased the number of CD8 positive cells in the tumors, while osimertinib and the osimertinib/sorafenib combination also enhanced the recruitment of proinflammatory macrophages, consistent with an immunosuppressive effect of EGFR signaling. We then compared the effects of these inhibitors on the growth of BEM-5 tumors in immunocompetent mice. While osimertinib and sorafenib alone slowed the growth of the tumors by about 2 weeks compared to the control, the combination induced an additional delay of almost 2-months (Fig. 7E, F). Together, our results indicate that co-treatment with sorafenib can substantially prolong the therapeutic response to osimertinib even in rapidly progressing tumors, and this effect may be enhanced by activation of an antitumor response by the immune system.

## Sorafenib inhibits the emergence of resistance from tolerant/persister cells

It has been shown that acquired resistance to osimertinib can either derive from pools of resistant cells already present within the tumors or from tolerant/persister cells capable of surviving during the treatment, which can then constitute a reservoir for the emergence of fully resistant cells[8–11]. In the studies described above, we used models based on pre-existing resistant clones. We next investigated the effects of the osimertinib-sorafenib combination on tolerant/persister cells. Our results showed that sorafenib can inhibit STAT3 phosphorylation in NSCLC cells. Intriguingly, it has been previously reported that increased STAT3 phosphorylation induced by targeted therapy can promote the survival of tolerant cells in different types of tumors, including NSCLC[57]. Consistent with these studies, we found that osimertinib stimulates STAT3 phosphorylation in PC9 cells, and this effect is inhibited by co-treatment with sorafenib (Fig. 8A), suggesting that co-treatment with this drug could also affect osimertinib-tolerant/persister cells. To test this hypothesis, we generated a pool of barcoded drug-tolerant expanded persisters (DTEPs)[58] (Supplementary Fig. 12A) by treating with osimertinib for 3 weeks a mass population of PC9 cells containing a control lentivirus, functioning as a barcode. These cells were then mixed with parental PC9 in a 1–100 ratio, followed by treatment with osimertinib, alone or in combination with sorafenib or crizotinib, used here as a negative control. As shown in Fig. 8B, osimertinib increased the proportion of DTEPs, and this effect was abolished by co-treatment with sorafenib, but not crizotinib, implying that this multikinase inhibitor can also inhibit tolerant/persister cells. We then used the Incucyte imaging system to investigate, through a different strategy, the effects of the osimertinib-sorafenib combination on the emergence of tolerant/persister cells. As expected, after massive cell kill by osimertinib, a population of surviving cells slowly emerged in the presence of this

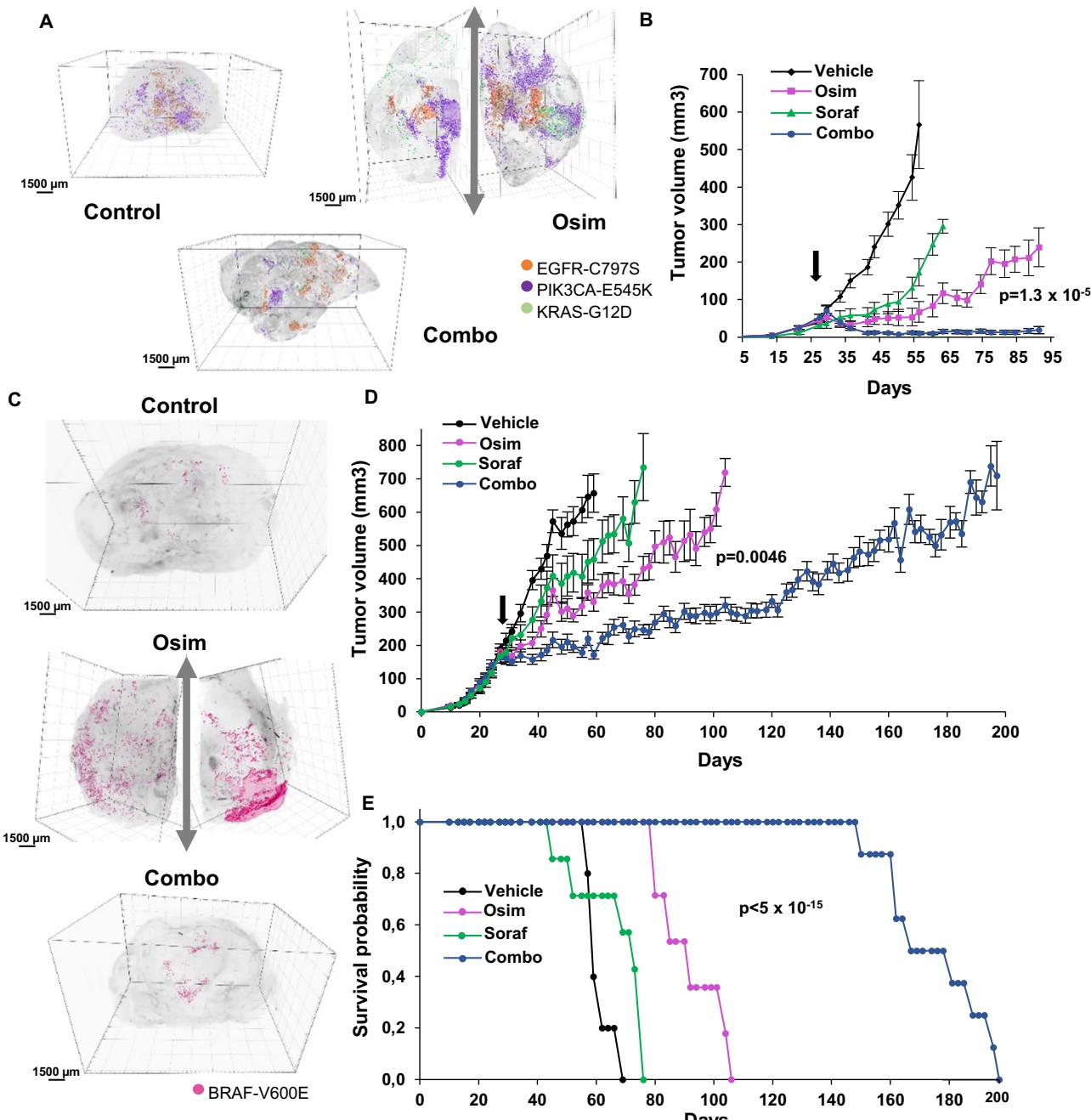

**Fig. 6 | The emergence of subpopulations of osimertinib-resistant NSCLC cells is inhibited in vivo by sorafenib. A** A PC9 mass population containing small pools (1:1000) of EGFR-C797S (expressing GFP), KRAS-G12D (expressing β-galactosidase), and PIK3CA-E545K (expressing mCherrry) osimertinib-resistant cells was subcutaneously injected in the flanks of female SCID mice. Once the tumors reached a mean volume of ~200 mm³, the mice were sacrificed (Ctrl) or treated for 4 weeks with osimertinib (5 mg/kg; Osim) alone or with sorafenib (60 mg/kg; Combo), followed by 3D-imaging. Images representative of two tumors per condition. The tumors from osimertinib-treated mice were cut in two. Scale bars: 1500 μm. **B** PC9 cells containing a small pool of EGFR-C797S cells were injected in the flanks of male SCID mice. Once the tumors were palpable (arrow), the mice were randomized and treated with or without osimertinib (5 mg/kg) and sorafenib (60 mg/kg; Soraf), alone or in combination, and tumor volume was measured by caliper. The mean tumor volumes ± SEM are represented (*n* = 5 mice). Osim vs Combo *p*-value was calculated at day 91 using two-tailed unpaired *t* test. **C** YUX-1024 PDCs containing a 1:200 subpopulation of BRAF-V600E resistant cells expressing GFP were subcutaneously injected in female SCID mice. Once the tumors reached a mean volume of about 200 mm³, the mice were sacrificed (Ctrl) or treated for 4 weeks with osimertinib (10 mg/kg) alone or in combination with sorafenib (60 mg/kg), followed by 3D-imaging. Images representative of two tumors per condition. Scale bars: 1500 μm. **D** YUX-1024 PDCs containing a pool of BRAF-V600E cells (1:200) were injected in the right and left flanks of female SCID mice. When the tumors reached a mean volume of about 200 mm³, the mice were treated with vehicle, osimertinib (10 mg/kg), sorafenib (60 mg/kg), or the combination. The mean tumor volumes ± SEM are represented (*n* = 6 mice for vehicle, *n* = 8 for the other groups); Osim vs Combo *p* value was calculated at day 83 using two-tailed unpaired *t* test. **E** Kaplan–Meier diagram of the experiment illustrated in (**D**). The mice were sacrificed when the volume of at least one of the tumors exceeded 800 mm³. Osim vs Combo *p* value was calculated (log-rank Mantel-Cox). Source data are provided as a Source Data file.

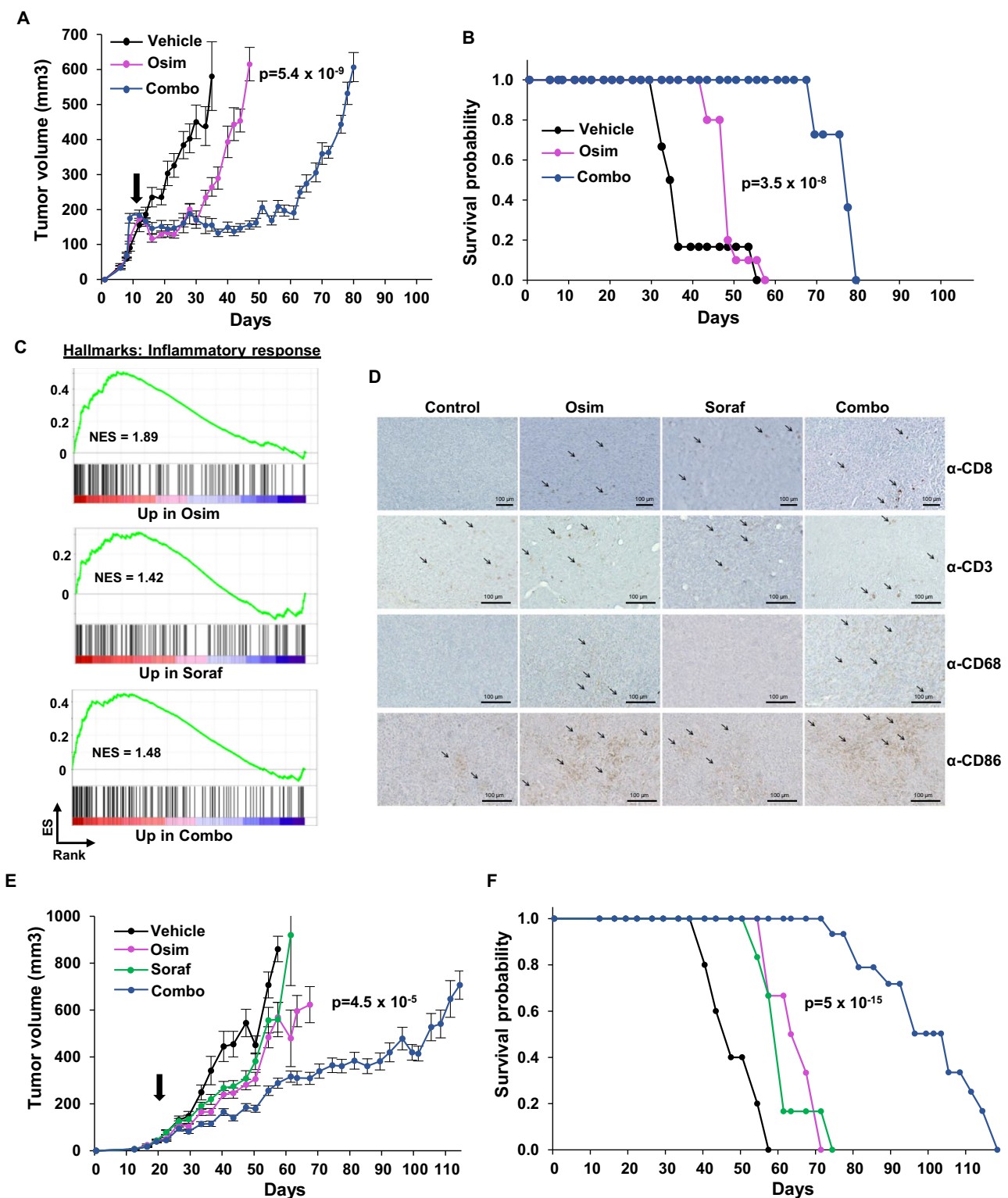

EGFR-TKI. Figure 8C and Supplementary Fig. 12B show that the growth of these tolerant/persister cells was strongly delayed by co-treatment with sorafenib.

By selecting cells that are less sensitive, drug treatment can provoke a major change in the clonal architecture of a cell mass population. To assess whether sorafenib could exert a broader inhibition of osimertinib-induced clonal evolution in NSCLC cells, we labeled several thousand PC9 clonal subpopulations using CRISPR highly complex barcodes[21] and compared the effects of a 2-week treatment with

osimertinib and sorafenib, either alone or in combination. We reasoned that pools of intrinsically tolerant/persister cells within the original population should be consistently enriched in the different replicates during the treatment, as opposed to randomly selected cells, which would be expected to emerge in only one or two replicates. As shown in Fig. 8D, the number of osimertinib intrinsically tolerant/ persister subpopulations was strongly reduced by co-treatment with sorafenib. Consistent with these data, the correlation plots depicted in Fig. 8E indicate that, while osimertinib induces a shift in the clonal

**Fig. 7 | The osimertinib-sorafenib combination promotes the recruitment of antitumor immune cells and substantially prolongs the response in extremely aggressive models of acquired resistance. A** A PC9 mass population containing small pools of CRISPR-barcoded EGFR-C797S, KRAS-G12D, and PIK3CA-E545K cells or ERBB2-ex20ins overexpressing cells was injected in the flanks of male and female SCID mice. Once the tumors reached a mean volume of about 100 mm³, the mice were randomized and treated with vehicle, osimertinib (5 mg/kg; Osim) alone or with sorafenib (60 mg/kg; Combo). The mean tumor volumes ± SEM are shown ($n = 6$ mice for vehicle, n = 10 mice for other groups). Osim vs Combo *p* value was calculated at day 47 using two-tailed unpaired *t* test. **B** Kaplan−Meier diagram of the experiment shown in (**A**). The mice were sacrificed when the volume of at least one of the tumors exceeded 800 mm³. Osim vs Combo *p* value was calculated (log-rank Mantel-Cox). **C** GSEA of the genes up-regulated by osimertinib, sorafenib, or the combination in PC9 cells based on an inflammatory response signature. **D** Mouse

BEM-5 cells were injected in the flanks of syngeneic BALB/c mice. When the tumors reached a mean volume of about 50 mm³, the mice were treated for 10 days with vehicle, osimertinib (20 mg/kg), sorafenib (60 mg/kg; Soraf), or the combination, followed by IHC analysis. Images representative of three different tumors/mice per condition. **E** BEM-5 cells were injected in the right and left flanks of BALB/c mice. Once the tumors reached a mean volume of about 50 mm³, the mice were randomized and treated with vehicle, osimertinib (20 mg/kg), sorafenib (60 mg/kg), or the combination. The mean tumor volumes ± SEM are shown ($n = 5$ mice for control, $n = 6$ mice for osimertinib and sorafenib, $n = 16$ mice for the combination). Osim vs Combo *p* value at day 57 was calculated using two-tailed unpaired *t* test. **F** Kaplan−Meier diagram of the experiment shown in (**E**). Osim vs Combo *p* value was calculated (log-rank Mantel-Cox). Source data are provided as a Source Data file.

architecture of the cancer cell population, this effect is inhibited by co-treatment with sorafenib.

To investigate whether inhibition of osimertinib-tolerant/persister cells by sorafenib could affect tumor growth, we used BEM-4 cells, another mouse clonal cell line expressing mutant EGFR we have generated (Supplementary Fig. 11D). When injected in BALB/c mice, these cells form tumors that initially respond to osimertinib but spontaneously relapse. Compared to osimertinib alone, the osimertinib-sorafenib combination further delayed tumor growth by about 3 weeks (Fig. 8F, G), indicating that co-treatment with sorafenib can improve tumor response to osimertinib even in the absence of a population of pre-existing resistant cells. Finally, to assess more in detail whether mice treated with the combination showed signs of stronger toxicity compared to those treated with osimertinib alone, we measured several blood parameters susceptible to be altered by treatment with these drugs, including lymphocytes, platelets, erythrocytes, and hemoglobin. Figure 8H shows that, for all the different parameters tested, the same values were obtained in mice treated with osimertinib or the combination. These data are consistent with the body weight measurements of the mice in all our in vivo experiments (Supplementary Fig. 13), indicating that the association with sorafenib has a low degree of toxicity in mice.

## Discussion

Clinically approved for two decades to treat NSCLC, EGFR-TKIs typically show high response rates and prolonged efficacy compared to other targeted therapies. Unfortunately, despite the strong addiction of NSCLCs to mutant EGFR and the high selectivity of these drugs for the mutant form of the receptor, these tumors almost inexorably relapse and become resistant to this treatment. To prevent this process, various strategies have been proposed to either achieve deeper inhibition of EGFR signaling or target a specific mechanism of tolerance/resistance[13–17,24,37,59,60]. However, complete inhibition of the pathway can cause unacceptable toxicity to normal tissues, and multiple mechanisms of tolerance/resistance generally coexist in the same patient. While acquired resistance to EGFR-TKIs in advanced NSCLC may be ineluctable, recent clinical trials have shown that co-treatment with chemotherapy can significantly delay progression[18–20].

Our present findings establish that, in combination with EGFR-TKIs, chemotherapy functions, at least in part, by inhibiting the growth of EGFR-TKI-resistant cells. Thus, these cells can be more vulnerable towards certain types of drugs because of their capacity to proliferate in the presence of EGFR-TKIs. By screening for other compounds that might be effective in inhibiting the emergence of cells with different mechanisms of resistance to EGFR-TKIs, we identified sorafenib and showed it to be specific for EGFR addicted cancer cells. Moreover, its effects could not be mimicked by other multikinase inhibitors. Importantly, though initially developed as a RAF inhibitor, we observed that the effects of sorafenib were mediated through different other intracellular pathways, rather than through inhibition of RAF

signaling. We showed that sorafenib rapidly blocked the catalytic activity of MKNKs, which can promote tumor progression by phosphorylating eIF4E, associated with efficient translation of several oncogenes[43,44], and through other mechanisms as well[61]. At later time points, we found that sorafenib inhibits STAT3 phosphorylation, as well as the expression of MCL1 and EGFR. Mechanistically, we showed that sorafenib did not inhibit the mRNA levels of MCL1 and EGFR, but it promoted instead the degradation of these two proteins through the proteasome and lysosome pathways, respectively. We also demonstrated that, while sorafenib similarly affected osimertinib-sensitive and resistant cells, it did not exert an additive effect with osimertinib. Our findings imply that, in a NSCLC population of cells containing both osimertinib-sensitive and resistant cells treated with the combination, osimertinib acts on sensitive cells, whereas sorafenib inhibits the growth of resistant cells (Supplementary Fig. 8). Thus, by relieving the selective pressure induced by osimertinib, sorafenib counteracts the capacity of the tumor cell population to adapt to treatment through clonal evolution.

Apart from fully resistant cells, it has been shown that certain cells capable of surviving in the presence of EGFR-TKIs can also play a role in the emergence of resistance. We found that the combination with sorafenib can inhibit these tolerant/persister cells both in vitro and in vivo. This effect is probably due, at least in part, to sorafenib-mediated repression of STAT3 phosphorylation, a transcription factor that can promote cancer cell survival upon EGFR inhibition[57]. It is also conceivable that the fact that tolerant/persister cells don't respond fully to EGFR-TKIs can represent a selective advantage, which can be targeted by co-treatment with sorafenib, as we demonstrated for EGFR-TKI-resistant cells.

Besides its approval for liver, thyroid, and kidney cancer, sorafenib has been tested in several other types of malignancy, including lung cancer. From a cohort of heavily pretreated patients, the BATTLE trial reported higher 8-week disease control rate for tumors containing wt EGFR. However, no significant differences in progression-free survival (PFS) were observed, and the study included only 12 patients with mutant *EGFR*[62,63]. The opposite conclusion was drawn from a larger phase III trial involving 89 and 258 patients displaying mutant and wt *EGFR*, respectively. Indeed, in the placebo-controlled MISSION study, sorafenib significantly improved both PFS (2.7 months for sorafenib versus 1.4 months for placebo) and OS (13.9 versus 6.5 months) in NSCLC patients with mutant *EGFR*, whereas no difference was observed when the whole cohort of patients was analyzed[28]. These data are consistent with our EGFR-score analysis and indicate that NSCLCs with activation of this receptor are more sensitive to sorafenib.

A phase II trial tested sorafenib in NSCLC patients after second- or third-line failure of first-generation EGFR-TKIs. While the authors didn't observe a significant correlation between tumor response and *EGFR* mutational status, the number of patients in this study was limited (22 patients with mutant and 8 patients with wt *EGFR*), and

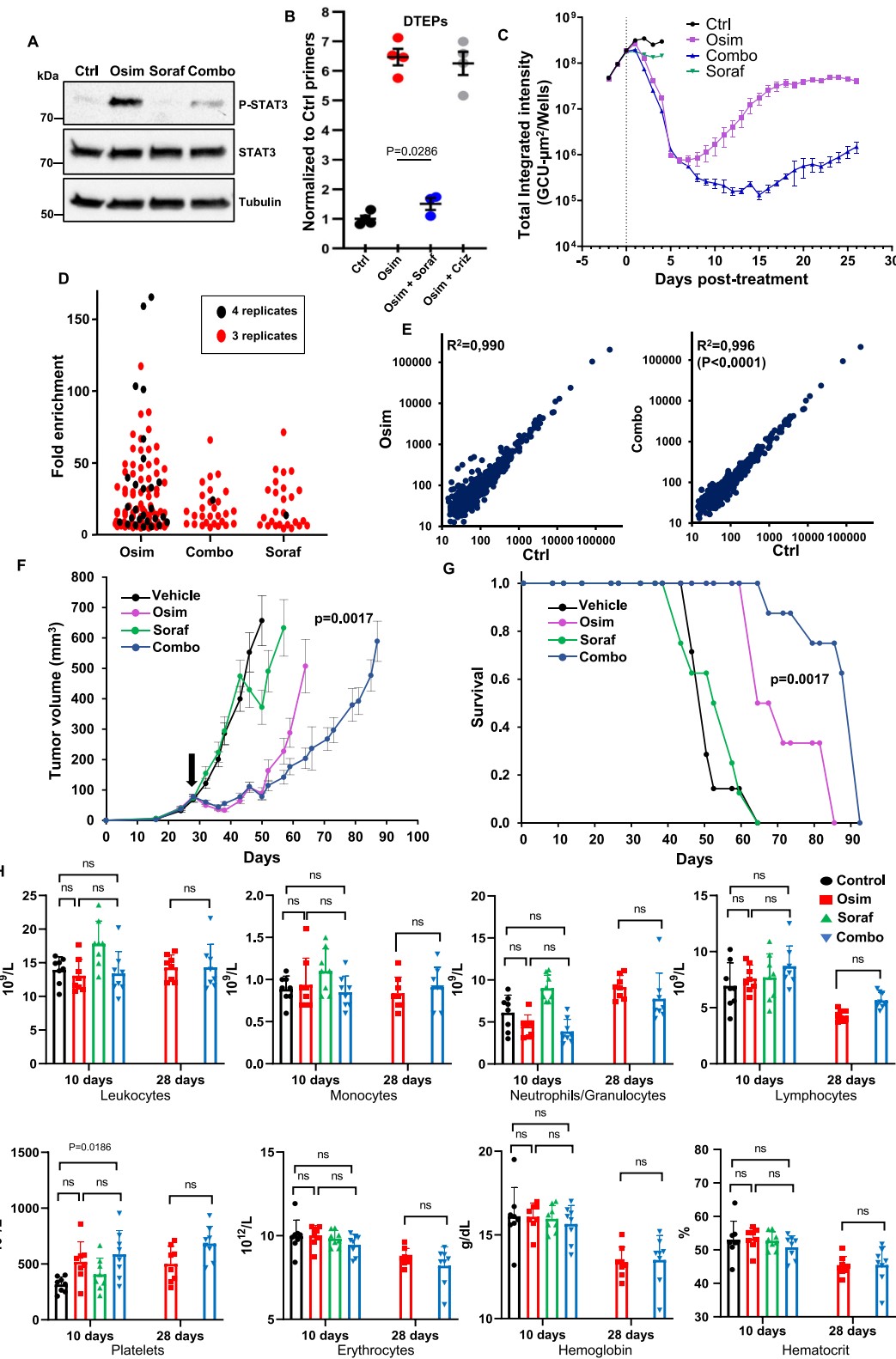

sorafenib was used as a single agent[64], which does not correspond to the sorafenib-osimertinib combination we propose here. A few studies have also tested the effects of sorafenib in combination with a first-generation EGFR-TKI in NSCLC patients. While some of these were only designed to evaluate the tolerability of the treatment[65,66], three multicenter phase II trials were performed to investigate the efficacy of the erlotinib-sorafenib combination. Two of these studies did not compare the effects of the combination with those of erlotinib alone, and the authors reported a better response in the few patients with mutant *EGFR* tumors[67,68]. The third trial found no difference between the combination and the EGFR-TKI alone, but it only involved five patients with mutant *EGFR* (two in the combination and three in the erlotinib group)[69]. While the very limited number of *EGFR*-mutant tumors in these cohorts precludes any meaningful evaluation of the potential

**Fig. 8 | Co-treatment with sorafenib inhibits osimertinib tolerant/persister cells. A** PC9 cells were treated for 1 d with osimertinib (1 μM; Osim) or sorafenib (5 μM; Soraf), alone or in combination (Combo), followed by immunoblot (representative of three independent experiments). **B** Drug-tolerant expanded persisters (DTEPs) labeled with a control lentivirus were mixed with parental PC9 cells (1:100) and treated for 15 d with osimertinib (0.1 μM) alone or with sorafenib (5 μM) or crizotinib (0.5 μM; Criz). DTEPs mean fraction ± SEM is shown (n = 4 biological replicates; representative of three independent experiments). Mann–Whitney two-tailed test. **C** PC9 cells stably expression GFP were treated with osimertinib (0,1 μM) or sorafenib (5 μM), alone or in combination for 26 d. The fluorescence from each well was measured every day with an Incucyte imaging system. The mean ± SEM of n = 7 biological replicates is shown (representative of two independent experiments). **D** PC9 cells containing highly complex CRISPR-barcodes in the AAVS1 locus were treated for 2 weeks with osimertinib (1 μM) and sorafenib (5 μM), alone or in combination (n = 4). Barcodes enriched at least 5-fold over the control in 4 or 3 biological replicates are shown. **E** Pearson correlation of the barcode distribution in control versus osimertinib or control versus combination. The coefficients of determination ($R^2$) and the p value (Mann–Whitney, two-tailed) are indicated. **F** Mouse BEM4 cells were injected in the flanks of syngeneic BALB/c mice. Once the tumors reached a mean volume of about 100 mm³, the mice were randomized and treated 3 times a week with vehicle, osimertinib (20 mg/kg), sorafenib (60 mg/kg), or the combination. The mean tumor volumes ± SEM are shown (n = 7 mice for the control, n = 8 mice for the other groups). Osim vs Combo p value at day 64 is shown (two-tailed unpaired t test). **G** Kaplan–Meier diagram of the experiment shown in (**F**). Osim vs Combo p value was calculated (log-rank Mantel-Cox). **H** The indicated blood parameters were measure from the mice in (**F**) at 10 d and 28 d after the beginning of the treatment. ns not significant (Kruskal–Wallis, two-tailed). Source data are provided as a Source Data file.

benefit of a sorafenib-EGFR-TKI combination, all these studies reported that the treatment was generally well tolerated, suggesting that the adverse effects of the association of osimertinib with sorafenib should also be manageable. Of note, these few early clinical trials are reminiscent of the several phase III studies that failed to prove the benefit of the combination of EGFR-TKIs with chemotherapy[70–74], before more recent studies demonstrated the efficacy of this strategy[18–20], which is now recommended by international treatment guidelines[75].

To clinically test a new drug combination, the expected benefits for the patients need to counterbalance the increased toxicity. Because the median progression-free survival of osimertinib as a single agent exceeds 18 months[2], the set-up of new clinical trials in treatment-naïve NSCLC patients can pose ethical concerns. To investigate the potential effects of our combination on more advanced tumors, we generated mouse models that mimicked rapid progression to EGFR-TKI treatment. We showed that co-treatment with sorafenib can substantially prolong the response of these aggressive tumors to osimertinib, indicating that the combination could also be beneficial during later phases of the disease.

Using a new syngeneic model of oncogenic addiction to mutant EGFR, we showed that the osimertinib-sorafenib combination is also effective in immunocompetent mice, where it promoted an inflammatory response. These findings are consistent with the immunosuppressive effects of EGFR signaling in NSCLC described in previous studies[53–55] and imply that the immune system could play a role in delaying the acquisition of osimertinib resistance. A contribution by the immune system might also explain the higher degree of variability in the individual response to the combination observed in immunocompetent versus immunodeficient mice (Kaplan–Meir diagrams of Fig. 7) and suggest that an analysis of the levels of infiltrated inflammatory cells induced by osimertinib may help to predict the duration of the response.

In conclusion, we showed that, similarly to chemotherapy, sorafenib can delay the acquisition of NSCLC resistance to EGFR-TKIs. While less efficient than EGFR-TKIs in inhibiting the growth of EGFR mutant cancer cells, sorafenib and chemotherapy specifically target the emerging subpopulations of resistant cells, thus prolonging NSCLC response to EGFR-TKIs. We showed that the osimertinib-sorafenib combination is well tolerated in mice even after several months of treatment, without any noticeable changes in all the different hematological parameters tested. According to the toxicities observed in the early clinical trials with first-generation EGFR-TKIs described above[67,68], sorafenib is expected to be more tolerable than chemotherapy in combination with osimertinib. Moreover, sorafenib can also affect tumor angiogenesis through VEGFR and PDGFR inhibition, and different studies have shown that co-treatment of EGFR-TKIs with antiangiogenic drugs can improve progression-free survival of NSCLC patients[76–78]. Altogether, our findings strongly support the clinical potential of combined osimertinib and sorafenib therapy for *EGFR* mutant NSCLCs.

## Methods

### Ethics statement
Animal experiments were approved by Mount Sinai's Animal Care and Use Committee (IACUC) or the French Regional Ethics Committees and the Ministry of Education, Research and Innovation (projects n°19253-201903191709966-v1− 16/10/2018 and 22757-2019110614311307−07/07/2020), and they were performed in accordance with the European Committee Council Directive (2010-63-EU).

### Cell culture and inhibitors
Human embryonic kidney 293T cells were obtained from ATCC; PC9 cells (NSCLC, EGFR-Ex19Del) were obtained from ECACC (distributed by Sigma-Aldrich); HCC4006 (NSCLC, EGFR-Ex19Del) and H1975 cells (NSCLC, EGFR-L858R/T790M) were obtained from ATCC; H3122 cells (NSCLC, EML4-ALK) were obtained from Cytion; HCC827 cells (NSCLC, EGFR-Ex19Del) and H358 (NSCLC, KRAS-G12C) were a gift from Pr. J. Minna, UT Southwestern Medical Center, Dallas (USA); HCC827-GR6 were a gift from Pr. P. Jänne, Dana-Farber Cancer Institute, Boston (USA); HCA-46 cells (CRC) were obtained from ECACC; CCK81 cells (CRC) were obtained from HSRRB (Japan); LIM1215 cells[79] were obtained from Prof. R. Whitehead, Vanderbilt University, Nashville (USA), with permission from the Ludwig Institute for Cancer Research, Zurich (Switzerland). NSCLC PDC lines YU-1150 (EGFR-L858R/T790M) and YUX-1024 (EGFR-L858R) were described previously[80]. BALB-3T3 cells were derived from BALB/c embryos[56]. The different cell lines were authenticated by short tandem repeat profiling (Eurofins Genomics). 293T cells were grown in DMEM (Life Technologies) supplemented with 10% FBS (Life Technologies); HCA-46 cells were grown in DMEM supplemented with 5% FBS; PC9, HCC4006, HCC827, H1975, YU-1150 and YUX-1024 were grown in RPMI1640 (2Mm L-glutamine +25 mM HEPES) supplemented with 10% FBS; LIM1215 cells were grown in RPMI1640 supplemented with 10% FBS, 0.5 μg/ml insuline (Sigma-Aldrich), 1 μg/ml hydrocortisone (Sigma-Aldrich) and 10 μM 1-thioglycerol (Sigma-Aldrich); CCK81 cells were grown in MEM (Life Technologies) supplemented with 10% FBS. BALB-3T3 cells were grown in DMEM supplemented with 10% calf serum (Life Technologies). All media were supplemented with 0.5% penicillin/streptomycin (Life Technologies). Drug-tolerant expanded persisters (DTEPs) were generated by growing PC9 cells in the presence of osimertinib (0.1 μM) for 3 weeks.

Osimertinib, sorafenib, cabozantinib, and lenvatinib were purchased from LC Laboratories. Gefitinib and doxycycline were purchased from Santa Cruz Biotechnology. Trametinib and puromycin were purchased from ChemCruz. Pemetrexed, sunitinib, eFT-508, napabucasin, S63845, and cetuximab were purchased from Selleckchem. Regorafenib was purchased from TargetMol. Crizotinib, chloroquine, and cycloheximide were purchased from Sigma-Aldrich. Sotorasib and MG132 were purchased by MedChemExpress. Blasticidin and zeocin were purchased from ThermoFisher.

## CRISPR-barcoding, cell transfection, and DNA constructs

sgRNA vectors were generated as previous described in ref. 21 by cloning annealed oligonucleotides containing the targeting sequence (Supplementary Table 1) into the pSpCas9(BB)−2A-Puro vector (a gift from Feng Zhang, Addgene plasmid # 48139)[81] digested with BbsI (New England Biolabs). The sequences of the donor single-stranded DNA oligonucleotide (ssODNs, Integrated DNA technologies) used for CRISPR/Cas9-mediated homology-directed repair, containing the mutation of interest and a few additional silent mutations constituting the barcode, are shown in Supplementary Table 2. Cells were co-transfected with 2 µg of the CRISPR/Cas9 plasmid and 2 µg of ssODN (50 µM) using a Nucleofector II and Amaxa Nucleofector kits (Lonza) according to the manufacturer's instructions. The efficiency of each transfection was checked in parallel using a GFP-containing plasmid.

Lentiviral vectors for inducible expression of CA-MKNK2 (T379D) and SNAI2 were generated by Gibson assembly (New England Biolabs) or standard cloning in the pTRIPZ plasmid from pTK-SNAI2 (a gift from Robert Weinberg, Addgene plasmid # 36986)[82] and pDONR223-MKNK2 (a gift from William Hahn and David Root, Addgene plasmid # 23735)[83]. Human dominant-negative MKNK1(T197A; T202A)[43] fused to GFP in N-terminus was cloned by Gibson assembly in the inducible VIPZ-puro lentiviral vector, originated from pTRIPZ and VIRHD plasmids. shMCL1 (targeting sequence: CGGGACTGGCTAGTTAAAC)[84] and shSTAT3 (targeting sequence: TGACTTTGATTTCAACTAT) were generated using the inducible lentiviral vectors VIPZ-zeo and VIPZ-bla, respectively.

The sequence encoding the Flag tag was cloned between the HindIII and KpnI sites of the pHA-eIF4E plasmid (a gift from Dong-Er Zhang, Addgene plasmid # 17343)[85]. Flag-tagged wt and Ex19Del human EGFR constructs in the lentiviral VIRSP vector were generated by PCR with Herculase II Fusion DNA Polymerase (Agilent Technologies) from EGFR WT (a gift from Matthew Meyerson, Addgene plasmid # 11011)[86] and pBabe EGFR Del1 (a gift from Matthew Meyerson, Addgene plasmid # 32062)[87] plasmids, respectively. The T2A peptide and EGFP were cloned by Gibson assembly downstream of EGFR-Ex19Del in the VIRSP plasmid. WT ERBB2 was cloned into VIRSP from the pCEV29-erbB2 plasmid previously generated by Makoto Igarashi in the Aaronson lab, and the sequence encoding A775insYVMA was added by site-directed mutagenesis and Gibson assembly. Mouse EGFR-L860R (corresponding to human EGFR-L858R) was cloned into VIRSP by PCR amplification from cDNA derived from a mouse hypothalamic cell line (Cellutions Biosystems) and Gibson assembly. For MET over-expression through CRISPRa[36], the lentiviral vectors dCAS-VP64_Blast, MS2-P65-HSF1_Hygro, and sgRNA(MS2)_zeo (gifts from Feng Zhang, Addgene plasmids # 61425, # 61426, and # 61427) were used. The MET sgRNA target sequence (5′-TGGCAGGGCAGCGCGCGTGT) was cloned in sgRNA(MS2)_zeo. To generate osimertinib-tolerant/persister cells, PC9 cells transduced with dCAS-VP64_Blast, MS2-P65-HSF1_Hygro, and empty sgRNA(MS2)_zeo were used. To label cells used in iDISCO 3D-imaging, lentiviral vectors GFP-VIRHD, mCherry-VIRHD, and pLenti-puro-LacZ (a gift from Ie-Ming Shih, Addgene plasmid # 39477)[88] were used. All constructs were sequence verified.

For lentivirus production, 293T cells were co-transfected using polyethylenimine (Polysciences) with the lentiviral vector, pCMV Δ8.91, and pMD VSV-G plasmids as previously described[89]. The conditioned media containing the viral particles were collected 2, 3, and 4 days after transfection, cleared by centrifugation, supplemented with 8 µg/ml polybrene, and added to PC9 cells for overnight incubation at 37 °C, 5% $CO_2$. Two days after transduction, the cells were selected in 1 µg/ml puromycin, 10 µg/ml blasticidin, or 250 µg/ml zeocin.

## DNA extraction, RNA extraction, and qPCR

Genomic DNA (gDNA) was extracted using NucleoSpin Tissue Kit (Macherey-Nagel) according to the manufacturer's instructions. Total RNA was isolated using the Tri-Reagent (Sigma-Aldrich) and chloroform (Sigma-Aldrich), purified with NucleoSpin RNA columns (Macherey-Nagel), quantified by Nanodrop One (Thermo Scientific), DNase digested, and reverse-transcribed using the Improm-II Reverse Transcription System (Promega). qPCR was performed from 100 ng of gDNA using the Fast SYBR Green Master Mix (Applied Biosystems) on a QuantStudio Flex PCR System (Thermo Scientific). The sequence of the different PCR primers, designed using Primer-BLAST (NCBI), is provided in Supplementary Table 3. For CRISPR-barcoding experiments, to avoid potential amplification from ssODN molecules not integrated in the correct genomic locus, one of the two primers was designed to target the endogenous genomic sequence flanking the region sharing homology with the ssODNs. qPCR analysis was performed using the standard curve method. For the initial CRISPR-barcoding screen, qPCR was performed using a QuantStudio 12 K Flex System (Thermo Scientific). The wt and mutant EGFR-T790 barcodes were detected by Taqman PCR using the EGFR-FW (5′-TCCCTCCAG-GAAGCCTACG) and EGFR-RV (5′-CCTTCCCTGATTACCTTTGCGA) primers and the EGFR-T790T (5′-CCAACTGATTACCCAGCTCATGCCC) and the EGFR-T790M (5′-GCTTATAATGCAACTGATGCCCTTCGG) probes, using a Taqman master mix (Thermo Scientific).

## Cell viability and colony-forming assays

For cell viability assay, 2500 PC9 cells per well were seeded in 96-well plates. Cells were incubated overnight to allow attachment to the plates, followed by treatment with the different drugs. The culture media/drugs were refreshed every 2/3 days. After 6 days of treatment, CellTiter-Glo reagent (Promega) was added to culture medium, mixed with an orbital shaker for 2 min to induce cell lysis, and incubated for 10 min at room temperature to stabilize the luminescent signal. Measurements were performed according to manufacturer's instructions using Infinite F200 PRO (TECAN). Cell viability was analyzed based on the levels of luminescence, proportional to the amount of ATP present in the cells.

For long-term colony-forming assays, 5000–10,000 cells per well were seeded in six-well plates. The following day, the cells were treated as indicated for the different experiments, then they were dried, fixed with a 10% MetOH, 10% acetic acid solution, and stained with a 1% crystal violet (Merck) MetOH solution. Excess stain was removed by rinsing in deionized water, and the plates were air-dried. Plate images were captured using a ChemiDoc Imaging System.

For long-term cell viability, PC9 cells stably expressing GFP were treated for more than 25 days with osimertinib (0.1 µM) and sorafenib (5 µM), alone or in combination, and washed with PBS at days 5 and 8. The cells were imaged every day, and the total fluorescence and confluency were quantified using the Incucyte® S3 Live-Cell Analysis System (Sartorius).

## EGFR-score analysis

We calculated EGFR pathway activity scores for each sample based on gene expression profiles, using a previously published EGFR gene signature[29]. This signature was derived from genes differentially expressed between EGFR-mutant and EGFR wild-type lung adenocarcinoma samples in The Cancer Genome Atlas (TCGA), integrating both transcriptomic and somatic mutation data. To quantify EGFR pathway activity, the signature was applied to our gene expression data using a rank-based scoring method[29], generating sample-specific EGFR scores, where higher scores reflect greater EGFR pathway activity in the corresponding samples. The EGFR score was calculated based on the gene expression data from patients with renal cell carcinomas[30] or thyroid cancer[31] treated with sorafenib (see the Data availability section).

## Flow cytometry

Cell cycle was analysed by flow cytometry analysis using the Click-iT EdU cell proliferation assay (ThermoFisher). Briefly, cells were labeled

with EdU (10 μM) for 1 h, followed by fixation, permeabilization, and staining with DAPI. The fluorescence was measured by flow cytometry using a CytoFLEX Flow Cytometer (Beckman Coulter), and the data were analysed using FlowJo software. To measure apoptosis, after trypsinization, the cells were centrifuged at $300 \times g$ for 5 min and washed twice in pre-cooled PBS. The cells were resuspended in 1x binding buffer and mixed with 2.5 μL Annexin V-FITC and PI by incubation in the dark at room temperature for 15 min. The fluorescence was measured by flow cytometry, and the data were analysed using FlowJo software.

In the experiment to investigate the effects of sorafenib on EGFR protein stability, parental or Flag-EGFR_Ex19Del-T2A-GFP (VIRSP)-transduced PC9 cells, treated with or without sorafenib (5 μM) for 3 days, were trypsinized, washed, and fixed with 4% paraformaldehyde. The levels of green fluorescence were measured by flow cytometry and analysed with the CytExpert software (Beckman Coulter).

### Immunoblot, m7GTP pull-down, and in vitro kinase assay

Cells were lysed in a buffer containing 50 mM HEPES pH 7.6, 150 mM NaCl, 5 mM EDTA, NP40 0.5%, 20 mM NaF, 2 mM Na$_3$VO$_4$, supplemented with protease inhibitor mini tablets (Thermo Scientific). Lysates were cleared by centrifugation at $14,000 \times g$ for 15 min at 4 °C, and protein concentration was determined using the Bradford assay (Bio-Rad). Sodium dodecyl sulfate (SDS) loading buffer was added to equal amounts of lysates, followed by SDS-polyacrylamide gel electrophoresis (PAGE) and transfer to polyvinylidene fluoride membranes (Bio-Rad) using a Trans-Blot Turbo Transfer System (Bio-Rad). Membranes were blocked for 1 h at room temperature in 5% nonfat dried milk (Sigma-Aldrich) in PBS and incubated overnight at 4 °C with the primary antibodies (Supplementary Table 4) in 3% BSA PBS-tween. After washing, the membranes were incubated 1 h with horseradish peroxidase-conjugated secondary antibodies (1:5000) and the bands were visualized by chemiluminescence using the Clarity Western ECL substrate (Bio-Rad) and a ChemiDoc Imaging System (Bio-Rad). The images were analyzed using the Image Lab Software (Bio-Rad).

To isolate cap-binding proteins, 40 μL of immobilized gamma-aminophenyl-m7GTP beads (Jena Bioscience) were added to 1 mg of PC9 cell lysates, followed by overnight incubation at 4 °C. After four washes in lysis buffer, SDS loading buffer was added to the beads, and immunoblot was performed in parallel from both the m7GTP-bound fraction and total lysate.

For in vitro MKNK2 kinase assay, 293T cells transfected with Flag-eIF4E (pcDNA3) were treated overnight with trametinib 0.1 μM, followed by cell lysis and immunoprecipitation using anti-Flag M2 agarose beads (Sigma-Aldrich). After extensive washing with cold lysis buffer and PBS, equal amounts of beads were incubated with kinase buffer (Abcam) supplemented with 300 μM ATP, 0.25 mM DTT and 50 μg/ml BSA, in the presence or the absence of 100 ng of recombinant human MKNK2 protein (Abcam) and different concentrations of sorafenib or other inhibitors for 20 min at 30 °C. The reaction was stopped by adding SDS loading buffer, followed by SDS-PAGE and immunoblot.

### Gene array analysis

PC9 cells were treated in triplicate with or without osimertinib (1 μM) and/or sorafenib (5 μM) for 1, 2 or 4 days (for the longest time point, the media was renewed after 2 days), and total RNA was isolated using the miRNeasy kit (Qiagen) following the manufacturer's instructions. The RNA quality was verified on agarose gel. Gene expression profiling was performed using a low-input QuickAmp labeling kit and human SurePrint G3 8x60K microarrays (Agilent Technologies), as previously described in ref. 90. Differentially expressed genes were identified by a 2-sample univariate t test and a random variance model, as previously described[91].

### Barcode sequencing

Highly complex CRISPR-barcodes were inserted into the AAVS1 locus of PC9 cells as previously described[21]. Barcoded cells were treated for 2 weeks with or without 1 μM osimertinib and 5 μM sorafenib, alone or in combination (four replicates per condition, one million cells per replicate in T-150 cm² flasks). After treatment, the cells were harvested and genomic DNA was extracted using NucleoSpin® Tissue kit (Macherey-Nagel) according to manufacturer's instructions. Targeted amplification of the integrated barcode in the AAVS1 locus was performed using previously described[21] primers containing Illumina adapter sequences and 6 bp unique indexes. For each sample, we performed 3 PCR reactions, each from 500 ng of genomic DNA in a final volume of 50 μl, using Herculase II Fusion DNA Polymerase (Agilent technologies) and the following program: 98 °C for 5 min; followed by 27 cycles of 20 s at 98 °C, 20 s at 60 °C and 30 s at 72 °C; final extension at 72 °C for 3 min. The PCR products from the same sample were pooled and purified over 2% agarose gels (band size at 275 bp) using NucleoSpin® Gel and PCR clean-up kit (Macherey-Nagel). Purified amplicons were quantified by Qubit (Thermofisher) and their quality was assessed by Bioanalyzer (Agilent). Sequencing was performed on an Illumina MiniSeq, using the High Output Reagent Kit (300-cycles). Counts of barcodes for each sample were extracted from FASTQ files using galaxy (https://usegalaxy.org). Heatmap and Ward's clustering analyses were performed using euclidean distance with R software. Spearman correlations were calculated using Excel (Microsoft).

### Generation of a syngeneic model of oncogenic addiction to mutant EGFR

BALB-3T3 cells were transduced with the lentiviral mouse EGFR-L860R vector and selected for 5 days in the presence of puromycin. To enrich for cells with stronger EGFR signaling activation, the cells were serum-starved over two cycles of 2 weeks. Addiction to EGFR-L860R was assessed by colony-forming assay in the presence of different concentrations of osimertinib. To select for tumorigenic cells, BALB-3T3-EGFR-L860R cells were bilaterally inoculated in the flanks of BALB/c mice. The tumors were dissociated, and the cells were grown in culture for 1 week in the presence of puromycin to eliminate the remaining cells from the host. Several clones were derived from the mass population by limiting dilution, and their reliance on EGFR signaling was assessed.

### Mouse xenografts

For the experiment illustrated in Fig. 6B, PC9 cells containing the EGFR-C797S CRISPR-barcode were mixed with Matrigel (Corning) and subcutaneously inoculated in the left and right flanks (2 × 106 cells per site) of 20 male SCID mice (6–8 weeks old). For the experiment shown in Fig. 7A, B, separate batches of PC9 cells containing the EGFR-C797S, KRAS-G12D, or PIK3CA-E545K CRISPR-barcodes were pooled and supplemented with a small fraction (1:200) of PC9 cells transduced with the ERBB2-ex20ins vector. The cells were mixed with Matrigel and bilaterally inoculated in 36 male and female SCID mice (6–8 weeks old). For the experiment depicted in Fig. 6D, E, YUX-1024 cells were transfected for BRAF-V600E CRISPR-barcoding and selected with 0.1 μM osimertinib for 3 weeks. The cells were then mixed with parental YUX-1024 in a 1:200 proportion, supplemented with 50% Matrigel, and bilaterally inoculated in the flanks of 40 female SCID mice (6–8 weeks). The size of the tumors was measured by caliper every 2–3 days. When tumors were palpable (Fig. 6B) or reached an average size of about 200 (Fig. 6D, E) or 100 mm³ (Fig. 7A, B) the mice were treated 3 times a week with osimertinib (5 mg/kg for experiments with PC9 cells; 10 mg/kg for experiments with YUX-1024 cells) and sorafenib (60 mg/kg), alone or in combination, using a water/EtOH/Kolliphor (Sigma-Aldrich) solution. For the experiment shown in Fig. 6B, the mice were initially treated 5 times a week for 12 days, before switching to a 3-times-a-week regimen for the rest of the experiment. When the size of at least one of

the tumors reached the arbitrary volume of 800 mm$^3$, the mice were sacrificed and the tumors were dissected for gDNA extraction.

For iDISCO 3D-analysis, PC9 cells containing the EGFR-C797S, PIK3CA-E545K or KRAS-G12D CRISPR-barcodes were selected with osimertinib (0.1 μM) and transduced with GFP, mCherry or β-galactosidase lentiviruses, respectively. The cells were selected with puromycin, and GFP and mCherry containing cells were sorted by FACS. The three cell populations were mixed with parental PC9 cells in a 1:1000 proportion to form a mass population of osimertinib-sensitive (unlabeled) and osimertinib-resistant (labeled) cells. YUX-1024 cells containing the BRAF-V600E CRISPR-barcode and selected with osimertinib (0.1 μM) were transduced with a GFP lentivirus, puromycin selected, and FACS sorted. The cells were then mixed with parental YUX-1024 (1:200 ratio). PC9 and YUX-1024 cells were mixed with Matrigel and subcutaneously inoculated in the flanks of 5 (YUX-1024) and 6 (PC9) female SCID mice (6–8 weeks old). When the tumors reached an average size of 200 mm$^3$, the mice were randomized and treated 3 times a week with osimertinib (5 mg/kg for the PC9 experiment, 10 mg/kg for the YUX-1024 experiment) alone or in combination with sorafenib (60 mg/kg) by gavage. Control mice were sacrificed after randomization. The other mice were sacrificed after 4 weeks of treatment, and the tumors were fixed overnight at 4 °C in 4% paraformaldehyde.

In the BEM5 allografts, a subpopulation of EGFR-C799S (corresponding to EGFR-C797S in the human receptor) cells was generated by CRISPR-barcoding BEM5 cells, a clone derived from BALB-3T3-EGFR-L860R cells. The cells were bilaterally injected in the flanks of 45 female BALB/c mice (6–8 weeks old; 2 × 106 cells per site). When the tumors reached an average size of 50 mm$^3$, the mice were treated 4 times a week with osimertinib (20 mg/kg) or 3 times a week with sorafenib (60 mg/kg), alone or in combination with osimertinib.

The BEM4 clonal cell line obtained from BALB-3T3-EGFR-L860R cells was injected in the right and left flanks of 32 syngeneic BALB/c mice (6–8 weeks old). Once the tumors reached a mean volume of about 100 mm3, the mice were randomized and treated with either vehicle or osimertinib (20 mg/kg), alone or in combination with sorafenib (60 mg/kg). At 10 and 28 days after the beginning of the treatment, blood was collected from the tails, and several blood parameters were measured using an Exigo H400 system (Fujifilm Vet Systems).

The mice were kept in a ventilated room at a temperature of 22 °C ± 0.5 °C under a 12-h light/12-h dark cycle (light on between 7:00 and 19:00) and 55% relative humidity. For all mouse experiments, the volume of the tumors was measured regularly to ensure that it did not exceed the maximal combined size of 1500 mm$^3$ permitted by our ethics committees.

## iDISCO 3D-imaging

The tumors were dehydrated with increasing concentrations of MetOH (20, 40, 60, 80, and 100%), followed by incubation in a solution of 2/3 dichloromethane (DCM; Sigma-Aldrich), 1/3 MetOH. They were then bleached overnight using a fresh 5% solution of $H_2O_2$ in MetOH at 4 °C, followed by rehydration with decreasing concentrations of MetOH (100, 80, 60, 40, and 20%). The tumors were incubated for 2 days with a permeabilizing solution containing PBS1, Triton-X 100, glycine, and 10% DMSO, then incubated for 2 additional days with a blocking solution containing 6% Donkey Serum, Triton-X 100, and 10% DMSO. The anti-GFP, anti-RFP/mCherry, and anti-beta-galactosidase antibodies (Supplementary Table 4) were incubated in PTwH/5%DMSO/3% Donkey Serum for 1 week at 37 °C. After 5 washes, the tumors were incubated with the secondary antibodies (Supplementary Table 4) in PTwH/3% Donkey Serum for 1 week at 37 °C. The tumors were washed 5 times and then dehydrated using increasing concentrations of MetOH, followed by a delipidation using DCM solutions and overnight clearing in dibenzylether (DBE; Sigma-Aldrich).

Cleared samples were imaged with an Ultramicroscope II (LaVision BioTec) using the ImspectorPro software (LaVision BioTec). The light sheet was generated by a Coherent Sapphire Laser (LaVision BioTec) at wavelengths of 488, 561, and 640 nm and six cylindrical lenses. A binocular stereomicroscope (MXV10, Olympus) with a 2x objective (MVPLAPO, Olympus) was used at different magnifications (0.63× and 0.8×). The samples were placed in an imaging reservoir made of 100% quartz (LaVision BioTec), filled with DBE and illuminated from the side by the laser light. Images were acquired with a PCO Edge SCMOS CCD Camera (2560 × 2160 Pixel size, LaVision BioTec). The step size in Z-orientation between each image was fixed at 4 or 6 μm for 0.63× and 0.8× magnifications. Images were processed and analyzed using the Imaris software (Bitplane).

## Immunohistochemical analysis of BALB/c tumors

Immunohistochemistry was performed on formalin-fixed and deparaffinized tissue sections. Sections were heated at 95 °C for 20 min in 10 mM citrate buffer (pH 6) and/or Tris EDTA (pH 9) for antigen retrieval, treated with peroxidase blocking reagent (Dako), and incubated with primary antibodies at the concentration indicated in Supplementary Table 4. The sections were incubated with secondary antibodies coupled to peroxidase and revealed with diaminobenzidine (Dako). The tissue sections were counterstained with hematoxylin, and images were acquired on an Axioscope 7 microscope (Zeiss).

## Statistics and reproducibility

Statistical analysis was performed with the GraphPad Prism software, and the tests used for each experiment are indicated in the legends. No statistical method was used to predetermine sample size. No data were excluded from the analyzes. The Investigators were not blinded to allocation during experiments and outcome assessment. The number of times each experiment has been independently performed is indicated in the Figure legends, and the results of each of those replicate experiments are provided in the Source Data File.

## Reporting summary

Further information on research design is available in the Nature Portfolio Reporting Summary linked to this article.

## Data availability

Gene expression data from renal cell carcinoma patients[30] were retrieved from Gene Expression Omnibus (GEO) GSE180925. RNA-seq data from thyroid cancer patients[31] were retrieved from BioProject (Submission ID: SUB6216503; BioProject ID:PRJNA563018) [http://www.ncbi.nlm.nih.gov/bioproject/563018]. The microarray datasets generated in this study were deposited into GEO (GSE179192) [https://www.ncbi.xyz/geo/query/acc.cgi?acc=GSE179192]. The remaining data are available within the Article, Supplementary Information, or Source Data file. Source data are provided with this paper.

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

## Acknowledgements

We thank Michael Thomas, Isabelle Lihrmann, Thierry Lecroq, Laurent Mouchard, Gilles Favre, and Céline Gongora for helpful discussion. We thank Pasi Jänne (Dana-Farber Cancer Institute), John Minna (UT Southwestern Medical Center), and Robert Whitehead (Vanderbilt University) for sharing cell lines. Some experiments and analyses, including part of the qPCR for the initial screen, amplicon quality assessment by Bioanalyzer, 3D-imaging analysis using the Imaris software, and acquisition of immunostained sections, were performed at the Cell Imaging Platform of Normandie (PRIMACEN). We thank Agathe Prieur (NorDiC), Gaëtan Riou (IRIB Flow Cytometry Facility), Julie Maucotel (SRB, UNIROUEN), Marine Di Giovanni, Magalie Bénard, and Alexis Lebon (PRIMACEN) for technical assistance. This work was supported by the Institut National de la Santé et de la Recherche Médicale (INSERM), the Université de Rouen Normandie, the Institut National du Cancer (PLBIO 2017-159; L.G.), the Agence Nationale de la Recherche (METROPOLIS-158530; A.C.), the Ligue contre le Cancer de Haute-Normandie (L.G.), the Fondation ARC pour la Recherche sur le Cancer (PJA20161205119; L.G.), the Conseil Régional de Normandie, the Contrat de Plan Etat-Région CPER Cancer 2015-2020 and the FEDER program of the European Union. L.B. was supported by a doctoral fellowship from the Ligue contre le Cancer de Normandie and the Normandie Region. A.G. was supported by a doctoral fellowship from the Normandie Region. H.C. was supported by doctoral fellowships from the Nabatieh Municipal Council (Lebanon) and the Fondation pour la Recherche Médicale. B.C.C. was supported by Basic Science Research Program through the NRF funded by the Korean Ministry of Science and ICT (2016R1A2B3016282). The research leading to these results has received funding from: the French Institut National du Cancer EU TRANSCAN23-002-2023-129, INCa_18688, PRT-K20-136 (C.Coulouarn); AIRC under IG 2023 -ID 29286 project (S.A.); FPRC 5×1000 Ministero della Salute 2022 CARESS (S.A.) and Italian Ministry of Health, Ricerca Corrente 2025 (S.A.); Prin 2022 PNRR financed by European Union - Next Generation EU M4 C2 I.1.1.-P2022E3BTH (S.A.); AIRC under 5 per Mille 2018 - ID. 21091 program (A.B.); AIRC under IG 2023 - ID. 28922 project (A.B.); PRIN 2022 - Prot. 2022CHB9BA financed by European Union - Next Generation EU (A.B.).

## Author contributions

L.B., D.A., J.L., MdM B.-R., D.B., A.G., H.C., M.G., Z.K., A.A., M.M., D.C., S.Y., D.G., J.D., P.J., A.V., and L.G. performed the experiments. L.B., D.A., J.L., MdM B.-R., D.B., A.G., H.C., Z.K., A.A., C.D., D.T., O.W., A.M., S.A.A., A.B.C., Y.A., and L.G. designed and/or analyzed experiments. J.-R.L. and C.Cheng analyzed the EGFR-score. C.B. participated in the analysis of highly complex CRISPR barcode. S.A. and A.B. provided CRC cell lines and participated in experiment design and analysis. C.Coulouarn performed and analyzed gene array experiments. B.C.C. provided the PDC lines and participated in experiment design and analysis. L.G., Y.A., and A.B.C. acquired funding. S.A.A., S.A., D.T., Z.K., Y.A., and L.G. edited the manuscript. L.G. conceived and supervised the study and wrote the manuscript.

## Competing interests

Z.K. reports financial support from DeuterOncology NV outside the submitted work. B.C.C. reports stock ownership with TheraCanVac Inc, Gencurix Inc, Bridgebio therapeutics, KANAPH Therapeutic Inc, Cyrus Therapeutics, Interpark Bio Convergence Corp and J INTS BIO; reports participating in an advisory role for KANAPH Therapeutic Inc, Brigebio Therapeutics, Cyrus Therapeutics, Guardant Health and Oscotec; has received consulting fees from Novartis, Abion, BeiGene, AstraZeneca, Boehringer-Ingelheim, Roche, Bristol-Myers Squibb, ONO, Yuhan, Pfizer, Eli Lilly, Janssen, Takeda, MSD, Janssen, Medpacto, and Blueprint medicines; has received grants or funds from Novartis, Bayer, AstraZeneca, MOGAM Institute, Dong-A ST, Champions Oncology, Janssen, Yuhan, ONO, Dizal Pharma, MSD, Abbvie, Medpacto, GI Innovation, Eli Lilly, Blueprint medicines, and Interpark Bio Convergence Corp; has received royalties from Campions Oncology, Crown Bioscience and Imagen; and is the founder of DAAN Biotherapeutics. A.B.C. has received honorarium for advisory positions, board memberships, lectures, or non-financial support from the following sources: Astra-Zeneca, Roche, MSD, Pfizer, Novartis, Takeda, Janssen, AbbVie, and Amgen. L.G. is inventor of a patent on DNA barcoding issued to Inserm and University of Rouen (WO2017068120A1). S. A. reports personal fees from MSD Italia and a patent (Italian patent application No. 102022000007535) outside the submitted work. A.B. reports receipt of grants/research supports from Neophore, AstraZeneca, Boehringer Ingelheim, and honoraria/consultation fees from Guardant Health. A.B. is stock shareholder of Neophore and Kither Biotech. A.B. is advisory boards member for Neophore. The remaining authors declare no competing interests.

## Additional information

[1]Univ Rouen Normandie, INSERM NorDiC UMR 1239, Rouen, France. [2]Institute for Research and Innovation in Biomedicine, Rouen, France. [3]Institut de Recherche en Cancérologie de Montpellier (IRCM), Inserm, Université de Montpellier, Institut Régional du Cancer de Montpellier (ICM), Montpellier, France. [4]Univ. Lille, CNRS, Inserm, CHU Lille, Institut Pasteur de Lille, UMR9020 – UMR1277 - Canther – Cancer Heterogeneity, Plasticity and Resistance to Therapies, Lille, France. [5]Univ Rouen Normandie, INSERM, CNRS, Normandie Université, HeRacLeS US51 UAR2026 SRB, Rouen, France. [6]Department of Oncological Sciences, Tisch Cancer Institute, Icahn School of Medicine at Mount Sinai, New York, NY, USA. [7]Department of Medicine, Baylor College of Medicine, Houston, TX, USA. [8]Institute for Clinical and Translational Research, Baylor College of Medicine, Houston, TX, USA. [9]Univ Rouen Normandie, LITIS EA 4108, Rouen, France. [10]Department of Oncology, University of Torino, Torino, TO, Italy. [11]Candiolo Cancer Institute, FPO - IRCCS, Candiolo, TO, Italy. [12]IFOM ETS - The AIRC Institute of Molecular Oncology, Milan, Italy. [13]Division of Medical Oncology, Yonsei Cancer Center, Yonsei University College of Medicine, Seoul, Republic of Korea. [14]Univ Rouen Normandie, INSERM, U1245, Cancer and Brain Genomics, Rouen, France. [15]Inserm, Univ Rennes, UMR_S 1242, Oncogenesis Stress Signaling (OSS) laboratory, Centre de Lutte contre le Cancer Eugène Marquis, Rennes, France. [16]Univ. Lille, CHU Lille, Thoracic Oncology Department, Lille, France. [17]These authors contributed equally: David Alexandre, Jiyoung Lee. ✉e-mail: luca.grumolato@univ-rouen.fr

