## [Transparent Peer Review file · Nature Communications]

Prolonging lung cancer response to EGFR inhibition by targeting the selective advantage of resistant cells

Corresponding Author: Dr Luca Grumolato

Version 0:

Reviewer comments:

Reviewer #1

(Remarks to the Author)

EGFR tyrosine kinase inhibitors (TKIs) have shown excellent efficacy for EGFR activating mutation-positive non-small cell lung cancer (NSCLC) and to improve the survival of individuals with this disease; however, acquired resistance to EGFR TKIs inevitably develops. In this manuscript, the authors used a DNA barcoding approach and showed that sorafenib, a multikinase inhibitor, abolished the selective advantage of EGFR-TKI-resistant cells and inhibited clonal evolution in vitro and in vivo. They also showed that the effects of sorafenib depends on early inhibition of MAPK interacting kinase (MNK) activity and signal transducer and activator of transcription 3 (STAT3) phosphorylation, and later down-regulation of MCL1 and EGFR, rather than RAF signaling inhibition. They also demonstrated that a combination of sorafenib and osimertinib recruited inflammatory cells in tumor areas. The authors conclude that a combination of osimertinib and sorafenib may delay the emergence of resistance in EGFR mutant NSCLCs.

Major

1. Overall: the major concern of this manuscript is that a combination of sorafenib and EGFR TKIs was evaluated in clinical trials more than ten years ago and it did not show impressive results to introduce this combination as an alternative strategy to overcome resistance to EGFR TKIs. The CTONG-0805 study evaluated the efficacy of sorafenib monotherapy as a second- or third-line therapy in patients with advanced lung adenocarcinoma after failure of EGFR-TKI therapy. However, the disease control rate had no significant difference between EGFR mutation-positive patients and EGFR wild-type patients (31.8% vs. 42.9%, respectively). In addition, the KCSG-0806 study shows that the median PFS is 11.0 months for patients with EGFR mutations treated with sorafenib and erlotinib, suggesting that sorafenib may not have a major impact on the emergence of resistance. Based on these results, it is unlikely that a combination of sorafenib and osimertinib will overcome resistance to osimertinib monotherapy in the clinical setting.
2. Fig.1A: the authors stated that “chemotherapy mainly affects the growth of cells that don’t respond to EGFR-TKIs”. However, this is an overstatement because pemetrexed clearly suppressed PC9 cell growth. This sentence should be rephrased.
3. Fig.1C: The authors state that “Together, these observations indicate that tumors with activation of EGFR signaling are sensitive to sorafenib,...” If this statement is true, why do they bother trying a combination strategy to begin with?
4. Extended Data Fig. 1C: use of more than 1 μM gefitinib may be unphysiological.
5. Fig.1F: Plasma concentrations of sorafenib seems to be lower in patients (Clin Cancer Res; 16(11) June 1, 2010). Thus, 5 μM in vitro may be a bit high if combination strategies are considered.
6. Fig.3: Sorafenib is a multi-targeted kinase inhibitor, reportedly inhibiting VEGFR-2, VEGFR-3, PDGFR-beta, c-Raf, b-Raf, c-Kit and Flt3. I wonder why the authors decided to focus on the Raf pathway over others?
7. Page 12, lines 303-305, and Fig. 3E-G and Fig. 5C: the authors stated that sorafenib prevents the emergence of EGFR-TKI resistant subpopulations of NSCLC cells through a mechanism involving, at least in part, its combined inhibitory effects on MNK, STAT3 and MCL1. However, this is an overstatement because Fig. 5C showed combination of osimertinib and sorafenib did not inhibit STAT3 phosphorylation in osimertinib-sensitive cells, indicating that sorafenib cannot prevent

emergence of resistant cells during osimertinib treatment. Also, the authors should show whether the osimertinib/sorafenib combination inhibits MNK activity and MCL1 expression in osimertinib-resistant, not just in osimertinib-sensitive cells.

8. Fig.3F: The mechanism underlying downregulation of MCL1 and MYC should be described. Would it be possible that protein degradation plays a role on it? In addition, if downregulation of these proteins is essential for sorafenib, this should be shown in *in vivo* experiments as well.

9. Extended Data Fig. 4E: the author should show that colony formation is suppressed by triple inhibition of inhibitors.

10. Fig.5C: Why is EGFR expression increased by osimertinib?

11. Fig.6 and 7: These *in vivo* experiments show that a combination of sorafenib and osimertinib suppresses intrinsic resistance, as tumor cells are pre-mixed before transplanted into mice. Thus it is impossible to evaluate if this combination inhibits emergence of drug-tolerant persister cells, which may be a dominant mechanism of acquired resistance.

Minor

1. p13, line 314: "didi" should read "did".

Reviewer #2

(Remarks to the Author)

The study by Brunet et al. uses a small-molecule screen to identify the pan-kinase inhibitor sorafenib as a therapeutic option for preventing the emergence of resistance to EGFR tyrosine kinase inhibitors in lung cancer. Using a CRISPR-mediated DNA barcoding approach that they previously developed (Guernet et al., *Mol Cell*, 2016), the authors performed a small-molecule screen to identify compounds that prevent the selection for cells with known resistance mechanisms following EGFR inhibition. They identified the multikinase inhibitor sorafenib as a hit in this screen and then characterize its function. They conclude that sorafenib can inhibit cells that are resistant to EGFR TKIs, and thereby prevent or delay resistance. They identify the kinase MNK as a likely target for sorafenib, and show that sorafenib inhibits Stat3 and leads to Mcl1 downregulation. They then perform a series of *in vitro* barcoding experiments and *in vivo* drug experiments to show that sorafenib inhibits the growth of TKI-resistant cells.

Overall this manuscript makes some interesting observations and uses a number of elegant approaches. However, there are several important weaknesses that need to be addressed. The most important is to clarify, both with new experiments and through the modifying text, the exact signaling context(s) in which sorafenib inhibits lung cancer proliferation and the mechanism(s) by which it acts.

Major concerns

1. The authors need to provide a much more thorough description of how sorafenib alone, in the absence of EGFR inhibitors, affects the proliferation and viability of the lung cancer models used in their paper. Sorafenib has previously been shown to inhibit the growth of lung cancer cells irrespective of their genotype (i.e. Kras mutant vs wild-type, or EGFR-mutant vs. wild-type; PMID 19638574), and the authors' findings are consistent with this (e.g. Figure 1I). Many of the findings in the current manuscript seem as though they can be explained by a model that sorafenib broadly inhibits the proliferation of lung cancer cells, independent of TKI sensitivity/resistance. Performing a detailed characterization of how sorafenib affects proliferation, survival, and relevant signaling pathways in EGFR-mutant lung cancer cells without co-treatment with EGFR TKIs is critical for understanding how this kinase inhibitor works to prevent TKI-resistance.
2. Related to Point #1 above, it is important to perform more detailed cellular characterization of how sorafenib affects proliferation and survival, both at baseline and following EGFR inhibition. There is evidence that drug-tolerant persister cells (DTPs) can exist in cycling or non-cycling states following EGFR inhibition (e.g. PMID 34381210). My understanding of the authors' model is that they would predict that sorafenib would inhibit the proliferation of cycling persister cells, but would not affect the survival or abundance of non-cycling persister cells. Examining the cellular consequences of sorafenib treatment would help strengthen their model.
3. The barcoding system employed is an elegant way to study the effects of a given pharmacologic perturbation on the selection for resistant cells following TKI treatment when these resistant cells are pre-existing in the cell population. However, TKI resistance can also emerge *de novo* during the persister cell state. It would be helpful to have a more explicit discussion of whether sorafenib affects the development of resistance arising during this stage.
4. Prior reports have shown that feedback activation of Stat3 following TKI treatment promotes TKI resistance (PMID 25065853). The authors should put their findings on the effect of sorafenib on Stat3 in the context of this prior work. For instance, in Figure 3E the authors show that sorafenib alone inhibits Stat3 phosphorylation. It would be important to demonstrate whether sorafenib also inhibits the feedback activation of Stat3 following EGFR inhibition.

Minor points

1. In Figure 1H, it is not clear that the combination of Osimertinib and cabozantinib is synergistic. Please consider softening this language or adding an experiment to directly assess synergy.
2. In their initial screen (Table S1) and in Figure 1K, sorafenib does not prevent the selection for Kras G12D following treatment of PC9 cells with gefitinib or the Kras G12C mutant cell line H358 with sotorasib, respectively. The authors conclude that sorafenib does not prevent the emergence of resistance in these settings. However, in Figure 2I, sorafenib does prevent the emergence of Kras G12D following treatment of the colon cancer cell line LIM1215 with cetuximab. The authors

conclude that sorafenib does prevent the emergence of resistance in this setting. However, these disparate results are confusing, since in all cases resistance is driven by Kras G12D, and it would seem as though sorafenib should have similar effects. The authors should clarify this, and address whether this implies that the effect of sorafenib is dependent upon the driving oncogene, and not the resistance mechanism.

3. In Figure 3E, please label the specific Stat3 phosphorylation being measured (Y705). Given that the section preceding this was discussing the serine/threonine kinase MNK, and Stat3 can also be phosphorylated on Serine 727, it is important to be precise so that the reader does not get the impression that MNK is directly responsible for the Stat3 Y705 phosphorylation. In fact, a brief discussion of what kinases mediate Stat3 phosphorylation might be helpful.

4. The authors state that the downregulation of Mcl1 and Myc are delayed (24 hours), but they do not show early time-points (1-2 hours) to demonstrate this.

5. For Figure 3H, it is important to show controls that the Mnk, Stat3, and Mcl1 inhibitors are working as described (e.g. pelf4E and pStat3; showing Mcl1 inhibition may be more difficult).

6. In Figure 5A, sorafenib alone induces widespread gene expression changes. It would be important to provide a description of these changes, e.g. by performing GSEA with Hallmarks gene sets.

7. The conclusion on Line 344 – 346 (These results ... suggest that, in a mixed cell population treated with both drugs, EGFR-TKI sensitive cells respond primarily to osimertinib, whereas sorafenib acts on resistant cells.) is confusing and does not seem to be supported by the data. The vast majority of cells in a bulk population of PC9 cells are sensitive to EGFR TKIs. Therefore it is hard to understand why sorafenib would induce such widespread gene expression changes (Figure 5A, right panel) and such a strong anti-proliferative response (Figure 1I) if it were only acting on TKI-resistance cells.

8. How were the Osimertinib-resistant cells in Figure 5C generated? Through CRISPR-mediated mutation, or spontaneous selection?

9. The data in Figure 6A and 6C should be quantified.

Reviewer #3

(Remarks to the Author)

The authors Brunet et al. present in their current manuscript the clonal outgrowth of resistant cells in NSCLCs during EGFR-TKI resistance using DNA-barcoding and counteracting this effect by employing a combination treatment with the multikinase inhibitor sorafenib.

While an intriguing concept is presented here and the effect of the combination treatment is demonstrated in xenograft and allograft mouse models, the authors showed a severe lack of inclusion of the proper experimental controls, replicate repeats and overall reporting of “n” numbers of their experiments. Specifically this includes the representation of one experiment (of a reported 3-4 independent repeats) in the following figures: Figure 1A,B,D,E,G,K; Figure 2A,B,D,E,G,H,I; Figure 3H; Figure 4F; Extended Data Fig. 1D; Extended Data Fig. 2A.

This is essentially showing an n of 1 for the majority of the data (listed above). The only figures that properly report on quantification of the data was found in extended data Fig 2B. Other experiments presented in the figures further fail to disclose the n numbers altogether, specifically: Fig. 1H, Fig. 2C, Extended Data Fig. 1A,B,F Extended Fig. 3A,D. Extended Fig. 5G (PCR quantification), Extended Data Fig 8A, Extended Data Fig 9B. Every immunoblot shown, additionally, has no reporting on the experimental n numbers at all in the respective figures. The following blots are missing the proper loading controls: Figure 3A-C, Extended Data Fig 3B and 3F. Figure 4A, Extended Figure 4A first blot, also the second and third do not report on total STAT3, Extended Data Figure 4B.

Furthermore, none of the colony formation assays are quantified nor an “n” number mentioned in the figure captions, namely: Figure 1F, 1I and 1L, Extended data Fig 1C & E, Extended data Fig 2 C+F, Extended data Fig 4E, some of which also seem to be acquired out of focus with phase contrast artefacts in the images, Extended Data Fig. 9A.

Therefore little to no conclusions can be drawn from a majority of the figures about the validity or reproducibility of the reported findings in the manuscript in its present form.

In order to make the data presented in the manuscript interpretable, authors would be requested to show the following: (1) individual data points in the bar graphs as well as the combination of the independent experimental repeats in these graphs, and NOT an n of 1 of a representative repeat (specific figures listed above). Additionally SEMs should be plotted as bot +/- in the graphs. (2) n numbers for the data presented as reported above, where there is currently no mention of them at all. (3) the appropriate loading controls of the immunoblots as well as annotations of ladder kDAs for each blot. (4) raw data of the immunoblots and their repeats. If there are no repeats of the immunoblots these will need to be performed and submitted. (5) quantifications of the colony formation assays, as well as again the n number repeats of these experiments.

Other major comments include the following:

Figure 6: Why were the Ctr mice sacrificed at 200mm² and the treatment mice after 4 weeks of treatment (i.e. at 800mm²)? In order to be truly comparable, mice should have been treated for the same time with vehicles and/or sacrificed when reaching a certain tumour size, i.e. 800 mm² for all experimental groups, not just the treated animals. There can be no reliable comparison of clonal size outgrowth to the control if it was already sacrificed at smaller volume/earlier and no conclusions drawn with regards to the increase in mutational clonal sizes if the controls are not allowed to reach the same size as the treated tumours. As a side note, the halving of the Osimertinib tumours, seems to be more a motivation to make the imaging more feasible than anything else, since no quantifications are shown (see minor comment below).

Figure 6D: An additive/cumulative tumour volume of both implanted flank tumours is plotted here for each animal or the volume of the larger of the two implanted tumours?

Minor comments:

Figure 2A: Why was the combination treatment of Osimertinib + Trametinib shown only for the PIKCA-E545K mutant and not for the others?

Page 13, line 312-314: The authors state: “[...] while other multikinase inhibitors, such as sunitinib and regorafenib, failed to effectively down-regulate the receptor (Extended Data Fig.5G).” This statement needs to be softened, since looking at the immunoblots, 5 µM regorafenib and sunitinib in the subsequent blot clearly do reduce the expression levels of the EGFR. Again, repeats of these experiments and quantifications would benefit the interpretation of this data.

Page 13, line 314, typo “didi” – did

Page 13, line 314-315: This statement: “Of note, we did not observe any effect of sorafenib on EGFR mRNA levels (Extended Data Fig.5G), [...]” cannot be made without any statistical analysis shown.

Figure 6A: The authors are asked to quantify the regions of the individual labelled mutations in order to make the claims these clones were enriched in the Osimertinib treated samples.

Figure 6A,C: The authors are requested to include the scale bar measurement in the figure legend.

Figure 6E, The authors are requested to plot any censored events in the Kaplan-Meier curve, as well as include relevant statistics in the figure, e.g. a log rank test between experimental groups.

Page 15 line 383-384: This statement: “The PDC line YUX-1024 forms tumors that are histopathologically similar to patient derived xenografts obtained from the same patient (Extended Data Fig.7B).” is just not true when examining the presented H&Es images, with the PDX F1 & 2 showing significantly enlarged nuclei and stronger stromal features compared to the PDC tumour, and this statement therefore needs to be softened.

Figure 7D, Please include scale bars in the figure.

Figure 7F, The authors are requested to plot any censored events in the Kaplan-Meier curve, as well as include relevant statistics in the figure, e.g. a log rank test between experimental groups.

Extended Data Fig.9E: This quantification should be moved to the main Figure 7 and shown with the necessary statistics in order to draw conclusions about the increase in certain immune populations.

Version 1:

Reviewer comments:

Reviewer #1

(Remarks to the Author)

The authors have provided additional data and addressed some of the reviewer’s questions. Studies using sorafenib including clinical trials were conducted more than ten years ago, and I certainly agree with their argument that previous clinical trials on the combination of EGFR-TKIs and sorafenib included only a small number of patients. However, the authors have not provided convincing results to overturn the findings that sorafenib may not have a major impact on the emergence of resistance to EGFR-TKIs. Another concern is that the manuscript does not provide mechanistic insights regarding effects of sorafenib on emergence of resistance to EGFR-TKIs. While they conclude that sorafenib prevents the emergence of EGFR-TKI resistant cells through the inhibition of MNK, STAT3, eIF4, and MCL1, each mechanism lacks in-depth investigation and the manuscript seems unfocused.

Major

Fig. 1F-I and Fig. 2: The authors showed that sorafenib can effectively prevent the emergence of resistance in cancer with EGFR mutation, but not in cells addicted to other oncogenic alterations such as KRAS and ALK in Fig. 1F-I. However, Fig. 2 shows that sorafenib is also effective against cells with resistance mechanisms independent on EGFR pathway signaling, including KRAS-G12D, BRAF-V600E, and EML4-ALK, in PC9 cells. It is not clear why sorafenib is effective in EGFR-mutated lung cancer with various resistance mechanisms independent of EGFR pathway signaling, despite poor efficacy against KRAS G12C or ALK fusion cell lines.

Fig. 1L and Fig. 5A: The authors showed that sorafenib alone does not affect cell survival in PC9 cells in Fig. 1L (page 9, line 195). However, Fig. 5A indicates that cell viability in PC9 parental cells after treatment with sorafenib alone is around 50%. Where does this discrepancy come from? Furthermore, cell viability in OR-PC9 cells after treatment with sorafenib alone was also around 50%. Despite PC9 parental cells having far fewer osimertinib-resistant clones than OE-PC9 cells, how are these results similar?

Fig.3: The authors conclude that sorafenib prevents the emergence of EGFR-TKI resistant cells through the inhibition of MNK, STAT3, and MCL1. However, no functional studies such as knockdown/knockout and overexpression have been conducted. Therefore, it remains unclear whether these inhibitions directly prevent the emergence of EGFR-TKI-resistant clones.

Fig. 3H: The reviewer does not understand what the authors question here. MCL1 downregulation was observed by CHX due to its inhibition of protein synthesis, and so did co-treatment with sorafenib. In this situation, the experiment does NOT imply that “sorafenib does not affect MCL1 protein synthesis” because MCL1 is already inhibited by CHX to begin with. Or do the authors mean that sorafenib may enhance downregulation of MCL1?

Fig. 8: In response to my comments regarding drug-tolerant persister cells, the authors evaluated the long-term effects in combination therapy in vitro and in vivo. However, they only evaluated PC9 cells which premixed with parental and DTEP cells in vitro. The reviewer is concerned that this is too artificial and not enough rationale for the 1:100 premix. No evaluation of characteristics of DTEP cells (e.g. sensitivity to osimertinib, which should be reversed) after labeling is provided. In vivo

evaluation of lung cancer cells is all based on PC9 cell line (BEM cells are fibroblasts). Lastly, adding other parameters than hematopoietic measurements for toxicity of the combination therapy would be beneficial.

Reviewer #2

(Remarks to the Author)

In the revised manuscript the authors address the majority of the issues raised in the initial review, and as a result the manuscript was significantly improved. I do not have any further concerns.

Reviewer #3

(Remarks to the Author)

The manuscript by Brunet et al. has greatly improved post rebuttal and the authors have gone to great length to show their work in the detail that is required for a reader audience to interpret the presented data fully. This reviewer is happy with the authors responses, having addressed the majority of this reviewers comments fully and using appropriate data presentation in their manuscript.

Version 2:

Reviewer comments:

Reviewer #1

(Remarks to the Author)

I thank the authors for their efforts to address my concerns. I do not have any further concerns.

Reviewer #4

(Remarks to the Author)

The comments of the Reviewers were very helpful, and we have performed a large number of additional experiments in response to their constructive suggestions. Our specific responses are detailed below:

Reviewer 1:

We appreciate Reviewer 1's constructive comments, and we have revised the manuscript in accordance with his/her specific suggestions as follows:

EGFR tyrosine kinase inhibitors (TKIs) have shown excellent efficacy for EGFR activating mutation-positive non-small cell lung cancer (NSCLC) and to improve the survival of individuals with this disease; however, acquired resistance to EGFR TKIs inevitably develops. In this manuscript, the authors used a DNA barcoding approach and showed that sorafenib, a multikinase inhibitor, abolished the selective advantage of EGFR-TKI-resistant cells and inhibited clonal evolution in vitro and in vivo. They also showed that the effects of sorafenib depends on early inhibition of MAPK interacting kinase (MNK) activity and signal transducer and activator of transcription 3 (STAT3) phosphorylation, and later down-regulation of MCL1 and EGFR, rather than RAF signaling inhibition. They also demonstrated that a combination of sorafenib and osimertinib recruited inflammatory cells in tumor areas. The authors conclude that a combination of osimertinib and sorafenib may delay the emergence of resistance in EGFR mutant NSCLCs.

Major:

1. Overall: the major concern of this manuscript is that a combination of sorafenib and EGFR TKIs was evaluated in clinical trials more than ten years ago and it did not show impressive results to introduce this combination as an alternative strategy to overcome resistance to EGFR TKIs. The CTONG-0805 study evaluated the efficacy of sorafenib monotherapy as a second- or third-line therapy in patients with advanced lung adenocarcinoma after failure of EGFR-TKI therapy. However, the disease control rate had no significant difference between EGFR mutation-positive patients and EGFR wild-type patients (31.8% vs. 42.9%, respectively). In addition, the KCSG-0806 study shows that the median PFS is 11.0 months for patients with EGFR mutations treated with sorafenib and erlotinib, suggesting that sorafenib may not have a major impact on the emergence of resistance. Based on these results, it is unlikely that a combination of sorafenib and osimertinib will overcome resistance to osimertinib monotherapy in the clinical setting.

The first clinical trials with sorafenib have been published more than 20 years ago and this drug has been tested in a very wide array of cancer types (a search in the ClinicalTrials.gov database using the keywords sorafenib and cancer resulted in 834 studies). Few of these trials are pertinent to our present study and we realized that we did not sufficiently discuss their relevance in the previous version of our manuscript. The phase II CTONG-0805 trial mentioned by the Reviewer (reference 64 of the revised manuscript) evaluated the efficacy of sorafenib alone after second- or third-line failure of first-generation EGFR-TKIs. This does not correspond to our proposed strategy based on combined sorafenib and osimertinib. As pointed out by the Reviewer, in this trial sorafenib was not more effective in patient with mutant compared to wt EGFR. However, this conclusion is based on only 22 patients with mutant EGFR and 8 patients with wt receptor. Also, only one of the 22 EGFR-mutant patients presented the EGFR-T790M mutation. This proportion is extremely low for tumors that developed resistance to erlotinib or gefitinib (the expected fraction would be around 50-60%), calling into question the reliability of the sequencing data reported in this study. A much larger phase III lung cancer study, involving 89 EGFR-mutant and 258 EGFR-wt patients,

showed improved PFS and OS in patients with mutant EGFR, but not in those with wt receptor (MISSION trial, reference 28). Consistent with these data, we analyzed the transcriptional signature of tumor samples in several published clinical trials and found that tumors with high EGFR-score responded better to sorafenib in various types of cancer, including lung, liver (reference 29) kidney (revised Fig.1F) and thyroid (new Fig.1G) cancer patients. Together, these observations suggest that tumors with activation of the EGFR signaling are more sensitive to sorafenib.

In the other phase II study cited by the Reviewer, KCSG-0806, the erlotinib-sorafenib combination was not compared to the erlotinib monotherapy and included only 8 patients with mutant EGFR tumors. The second author of the paper describing this trial (reference 68) is also a co-author of our present manuscript. The two other clinical studies that tested the effects of the erlotinib-sorafenib combination (references 67 and 69) also included very few patients with mutant EGFR (7 and 2, respectively), and many had also received previous cycles of chemotherapy. In view of the strong positive pre-clinical findings in our present manuscript, it seems fair to conclude that the combination of EGFR-TKIs with sorafenib has not been subjected to an appropriate clinical test. This is reminiscent of the combination of EGFR-TKIs with chemotherapy, which failed in several large phase III trials (including INTACT 1, INTACT 2, OSI-774 and FASTACT-2, references 70-74), but showed significant benefit in PFS and OS in recent phase III studies with a more appropriate design (references 19-20). Of note, the first-line association of osimertinib with chemotherapy is now recommended by international treatment guidelines (reference 75). Together, our present findings provide a strong rationale for a clinical trial that would conclusively assess the efficacy of the osimertinib-sorafenib combination in an appropriate cohort of EGFR-mutant patients. We have clarified these points in the revised discussion.

2. Fig.1A: the authors stated that “chemotherapy mainly affects the growth of cells that don’t respond to EGFR-TKIs”. However, this is an overstatement because pemetrexed clearly suppressed PC9 cell growth. This sentence should be rephrased.

We apologize for the misunderstanding. We didn’t imply that EGFR-TKI-resistant cells are more sensitive to pemetrexed compared to EGFR-TKI-sensitive cells. What we meant was that, in the presence of the gefitinib-pemetrexed combination, gefitinib inhibits the growth of gefitinib-sensitive cells, which are then unable to respond to pemetrexed, since they stopped cycling. On the contrary, gefitinib-resistant cells, which, by definition, are not affected by this EGFR-TKI, can be inhibited by pemetrexed. This model is supported by the lack of additive effect of the two drugs (Fig.1A), as well as our barcoding experiments (Fig.1B). We have rephrased the sentence to clarify this point.

3. Fig.1C: The authors state that “Together, these observations indicate that tumors with activation of EGFR signaling are sensitive to sorafenib,...” If this statement is true, why do they bother trying a combination strategy to begin with?

Our analysis of the transcriptional profile of tumor samples from several clinical studies revealed that tumors with high EGFR score are more likely to respond to sorafenib treatment (reference 29, Fig.1F and new Fig.1G), which is consistent with the results of the MISSION trial discussed above. This is also consistent with our mechanistic studies, showing that sorafenib induces down-regulation of EGFR. We believe that these observations can support the association of sorafenib with osimertinib to delay the emergence of resistance, but we definitely don’t argue that sorafenib could be used to replace osimertinib or other EGFR-TKIs. The sentence cited by the Reviewer has been toned down in the revised manuscript.

4. *Extended Data Fig. 1C: use of more than 1 μM gefitinib may be unphysiological.*

Agree. We have modified the figure according to the Reviewer's suggestion.

5. *Fig.1F: Plasma concentrations of sorafenib seems to be lower in patients (Clin Cancer Res; 16(11) June 1, 2010). Thus, 5 μM in vitro may be a bit high if combination strategies are considered.*

The concentrations of sorafenib in both our *in vitro* and *in vivo* experiments are those typically used in the literature. Pharmacokinetic studies showed that sorafenib has a very high protein binding (reference 23). Comparison of its total and unbound fractions in both plasma and cell culture media revealed that the most commonly used *in vitro* concentrations (between 3 and 10 μM) are similar to what can be achieved in patients treated with the approved dose of 400 mg twice a day (reference 22). Consistent with these observations, while all our *in vitro* experiments were performed in 10% FBS media, we noticed that at lower serum concentrations, which correspond to lower protein levels, the effects of sorafenib were much stronger (data not shown). More importantly, our *in vivo* studies showed that the osimertinib-sorafenib combination is effective in delaying tumor relapse and can be used over long periods, without any sign of toxicity (new Fig.8H).

6. *Fig.3: Sorafenib is a multi-targeted kinase inhibitor, reportedly inhibiting VEGFR-2, VEGFR-3, PDGFR-beta, c-Raf, b-Raf, c-Kit and Flt3. I wonder why the authors decided to focus on the Raf pathway over others?*

We initially focused on the RAF pathway because we found in early experiments that sunitinib, which targets the same tyrosine kinases than sorafenib, but not the serine/threonine kinase RAF, could not mimic the effects of sorafenib in blocking the emergence of EGFR-TKI resistant cells. Also, given the major role of MAPKs in EGFR signaling, we thought that RAF was an obvious candidate.

7. *Page 12, lines 303-305, and Fig. 3E-G and Fig. 5C: the authors stated that sorafenib prevents the emergence of EGFR-TKI resistant subpopulations of NSCLC cells through a mechanism involving, at least in part, its combined inhibitory effects on MNK, STAT3 and MCL1. However, this is an overstatement because Fig. 5C showed combination of osimertinib and sorafenib did not inhibit STAT3 phosphorylation in osimertinib-sensitive cells, indicating that sorafenib cannot prevent emergence of resistant cells during osimertinib treatment. Also, the authors should show whether the osimertinib/sorafenib combination inhibits MNK activity and MCL1 expression in osimertinib-resistant, not just in osimertinib-sensitive cells.*

We apologize for not making sufficiently clear our model for the mechanism of action of sorafenib. We showed that sorafenib inhibits MNK activity, STAT3 phosphorylation and the expression of MCL1 and EGFR. The same effects can be observed in both EGFR-TKI-sensitive and resistant cells (new Fig.5C, Fig.5D, Supplementary Fig.6B) in the presence of sorafenib alone and they are consistent with the growth inhibition induced in these cells (Fig.5A-B and Supplementary Fig.6A). These effects can also be observed in the presence of the osimertinib-sorafenib combination in osimertinib-resistant cells, since osimertinib has no effect in these cells, which can thus only respond to sorafenib. Conversely, in osimertinib-

sensitive cells, while MCL1 is still inhibited, EGFR and STAT3 are not (new Fig.5C, Fig.5D, new Fig.8A). As for MNK activity, it can be inhibited by both sorafenib and osimertinib, albeit through a different mechanism (we showed that sorafenib directly inhibits MNK catalytic activity, while osimertinib acts indirectly through the EGFR-MAPK-MNK pathway). These findings are consistent with the lack of additive effects by the two drugs (Fig.5A-B) and are confirmed by our GSEA analysis of their transcriptional effects (Fig.5E-F), implying that, in osimertinib-sensitive cells, the effects of sorafenib are largely inhibited by co-treatment with osimertinib. Therefore, according to our model (Supplementary Fig.8), in a mixed population containing both osimertinib-sensitive and osimertinib-resistant cells, treatment with the osimertinib-sorafenib combination results in growth inhibition of sensitive cells induced by osimertinib (since in these cells the effects of sorafenib are countered by the presence of osimertinib) and growth inhibition of resistant cells induced by sorafenib (since these cells are unresponsive to osimertinib). As a consequence, co-treatment with sorafenib inhibits the selective advantage of osimertinib-resistant cells, thus repressing their amplification. We clarified this important point in revised Fig.5 and the corresponding section of the results, and the model is illustrated by the diagram in revised Supplementary Fig. 8. Also, as requested by the Reviewer, we have tested the effects of sorafenib on MNK activity and MCL1 expression, alone or with osimertinib, in both osimertinib-sensitive and resistant cells (new Fig.5C).

8. Fig.3F: The mechanism underlying downregulation of MCL1 and MYC should be described. Would it be possible that protein degradation plays a role on it? In addition, if downregulation of these proteins is essential for sorafenib, this should be shown in in vivo experiments as well.

Agree. We performed additional experiments that revealed that sorafenib doesn't inhibit MCL1 mRNA levels (new Supplementary Fig.4E) nor MCL1 protein synthesis (new Fig.3H), but it promotes instead MCL1 degradation through the proteasome pathway (new Fig. 3I). Also, our new *in vivo* experiments showed that sorafenib treatment can inhibit EGFR and MCL1 expression in tumors (new Fig.4G). Since the effects on MYC didn't represent a major component of the mechanism of sorafenib in NSCLC cells, we preferred not to include these results in the revised manuscript.

9. Extended Data Fig. 4E: the author should show that colony formation is suppressed by triple inhibition of inhibitors.

Agree. We performed the experiment suggested by the Reviewer, which is now represented in Supplementary Fig.4F.

10. Fig.5C: Why is EGFR expression increased by osimertinib?

In some experiments we noticed that EGFR-TKI treatment, while inhibiting EGFR phosphorylation, can also induce upregulation of this receptor in PC9 cells. This is probably due to a compensatory mechanism, which are common in RTK pathways. For example, it has been shown that MAPK inhibition can result in feedback activation of EGFR in colon cancer cells (Prahallad et al., Nature 483). Further studies would be needed to characterize the underlying mechanism, but we believe this would be beyond the scope of our manuscript.

11. Fig.6 and 7: These in vivo experiments show that a combination of sorafenib and osimertinib suppresses intrinsic resistance, as tumor cells are premixed before transplanted

into mice. Thus it is impossible to evaluate if this combination inhibits emergence of drug-tolerant persister cells, which may be a dominant mechanism of acquired resistance.

We thank the Reviewer for the very useful suggestion. We performed a series of new *in vitro* and *in vivo* experiments, depicted in the new Figure 8, which showed that co-treatment with sorafenib can also inhibit osimertinib tolerant/persister cells.

Minor

1. p13, line 314: “didi” should read “did”.

Agree. We corrected the typo.

Reviewer 2:

We appreciate Reviewer 2's overall positive comments and we have addressed the weaknesses he mentioned by performing several new experiments as indicated below:

The study by Brunet et al. uses a small-molecule screen to identify the pan-kinase inhibitor sorafenib as a therapeutic option for preventing the emergence of resistance to EGFR tyrosine kinase inhibitors in lung cancer. Using a CRISPR-mediated DNA barcoding approach that they previously developed (Guernet et al., Mol Cell, 2016), the authors performed a small-molecule screen to identify compounds that prevent the selection for cells with known resistance mechanisms following EGFR inhibition. They identified the multikinase inhibitor sorafenib as a hit in this screen and then characterize its function. They conclude that sorafenib can inhibit cells that are resistant to EGFR TKIs, and thereby prevent or delay resistance. They identify the kinase MNK as a likely target for sorafenib, and show that sorafenib inhibits Stat3 and leads to Mcl1 downregulation. They then perform a series of in vitro barcoding experiments and in vivo drug experiments to show that sorafenib inhibits the growth of TKI-resistant cells.

Overall this manuscript makes some interesting observations and uses a number of elegant approaches. However, there are several important weaknesses that need to be addressed. The most important is to clarify, both with new experiments and through the modifying text, the exact signaling context(s) in which sorafenib inhibits lung cancer proliferation and the mechanism(s) by which it acts.

Major points

1. The authors need to provide a much more thorough description of how sorafenib alone, in the absence of EGFR inhibitors, affects the proliferation and viability of the lung cancer models used in their paper. Sorafenib has previously been shown to inhibit the growth of lung cancer cells irrespective of their genotype (i.e. Kras mutant vs wild-type, or EGFR-mutant vs. wild-type; PMID 19638574), and the authors' findings are consistent with this (e.g. Figure 11). Many of the findings in the current manuscript seem as though they can be explained by a

model that sorafenib broadly inhibits the proliferation of lung cancer cells, independent of TKI sensitivity/resistance. Performing a detailed characterization of how sorafenib affects proliferation, survival, and relevant signaling pathways in EGFR-mutant lung cancer cells without co-treatment with EGFR TKIs is critical for understanding how this kinase inhibitor works to prevent TKI-resistance.

Agree. We have performed new experiments to investigate the effects of sorafenib on cell cycle and cell survival. As shown in the new Fig.1K and 1L and the new Supplementary Fig.1E and 1F, sorafenib inhibits the proliferation of both PC9 and HCC827 cells, while it also promotes apoptosis in HCC827 cells. In our manuscript we demonstrated that sorafenib acts through inhibition of MNK catalytic activity, STAT3 phosphorylation, and MCL1 and EGFR expression. We investigated the effects of sorafenib in combination with EGFR-TKIs (revised Figure 5, which includes two new panels). These studies revealed that some of the effects that sorafenib exerts when it is used as single agent are blocked in the presence of the combination, including the inhibition of STAT3 phosphorylation and EGFR expression. We also showed that both sorafenib and EGFR-TKIs can repress MNK activity, albeit through a different mechanism (through direct inhibition of its catalytic activity for sorafenib, through inhibition of the EGFR-MAPK-MNK pathway for EGFR-TKIs). Moreover, we found that growth inhibition induced by sorafenib and EGFR-TKIs are not additive (Fig.5A-B and Supplementary Fig. 6A) and the transcriptional profile of cells treated with the osimertinib-sorafenib combination is close to that of cells treated with osimertinib alone (Fig.5E-F and Supplementary Fig.7A-C). Together, all these findings imply that, in a mixed population containing both osimertinib-sensitive and osimertinib-resistant cells, treatment with the combination provokes the inhibition of sensitive cells by osimertinib (since in these cells the effects of sorafenib are restrained by the presence of osimertinib) and the inhibition of osimertinib-resistant cells by sorafenib (since these cells are, by definition, unresponsive to osimertinib). Therefore, the end result of sorafenib co-treatment is the inhibition of the selective advantage of osimertinib-resistant cells, which can no longer be amplified in the presence of osimertinib, as we demonstrated with our barcoding approach. The Reviewer is correct that sorafenib can also affect the growth of NSCLC cells addicted to other oncogenes, such as H358 (mutant KRAS) and H3122 (ALK mutant) cells. However, we showed that in these cells the inhibitory effects of sorafenib are additive with those of targeted therapy (new Fig.5B). Thus, when these cells are co-treated with sorafenib and targeted therapy, sorafenib can affect the growth of both sensitive and resistant cells to a similar extent. As a consequence, in NSCLC cells not addicted to EGFR, sorafenib is unable to inhibit the selective advantage of resistant cells, as we demonstrated in H358 and H3122 cells treated with sotorasib (KRAS inhibitor) or ceritinib (ALK inhibitor), respectively (Fig.1H and new Fig.1I). Our model is described with a diagram in Supplementary Fig.8 and we have clarified this point in the revised Fig.5 and the related paragraph of the results.

2. Related to Point #1 above, it is important to perform more detailed cellular characterization of how sorafenib affects proliferation and survival, both at baseline and following EGFR inhibition. There is evidence that drug-tolerant persister cells (DTPs) can exist in cycling or non-cycling states following EGFR inhibition (e.g. PMID 34381210). My understanding of the authors' model is that they would predict that sorafenib would inhibit the proliferation of cycling persister cells, but would not affect the survival or abundance of non-cycling persister cells. Examining the cellular consequences of sorafenib treatment would help strengthen their model.

As mentioned in the response to the previous point, sorafenib inhibits the proliferation of NSCLC cells. However, co-treatment with EGFR-TKI impairs this effect in EGFR-TKI

sensitive cells, thus in the combination settings sorafenib mainly acts on EGFR-TKI-resistant cells, preventing their selection. We performed a series of new experiments to investigate the effects of sorafenib co-treatment on tolerant/persister cells (please see our response to the next point of the Reviewer).

3. The barcoding system employed is an elegant way to study the effects of a given pharmacologic perturbation on the selection for resistant cells following TKI treatment when these resistant cells are pre-existing in the cell population. However, TKI resistance can also emerge de novo during the persister cell state. It would be helpful to have a more explicit discussion of whether sorafenib affects the development of resistance arising during this stage.

Agree. We thank the reviewer for this suggestion. We performed several additional experiments, both *in vitro* and *in vivo*, to investigate the effects of the osimertinib-sorafenib in models that did not contain pre-mixed resistant cells. As shown in the new Figure 8, we found that co-treatment with sorafenib can not only repress resistant cells, but also inhibit the emergence of tolerant/persister cells.

4. Prior reports have shown that feedback activation of Stat3 following TKI treatment promotes TKI resistance (PMID 25065853). The authors should put their findings on the effect of sorafenib on Stat3 in the context of this prior work. For instance, in Figure 3E the authors show that sorafenib alone inhibits Stat3 phosphorylation. It would be important to demonstrate whether sorafenib also inhibits the feedback activation of Stat3 following EGFR inhibition.

Agree. We are grateful to the Reviewer for this very useful comment. As he/she suggested, we tested whether sorafenib co-treatment can affect STAT3 activation by osimertinib. The new Fig.8A shows that osimertinib stimulated STAT3 phosphorylation, which, as pointed out by the Reviewer, has been shown to promote the survival of cancer cells in the presence of targeted therapy (the paper cited by the Reviewer has been added as reference 57). We found that sorafenib co-treatment can inhibit this effect, which is probably part of the mechanism by which this drug can counter the emergence of tolerant/persister cells.

Minor points

1. In Figure 1H, it is not clear that the combination of Osimertinib and cabozantinib is synergistic. Please consider softening this language or adding an experiment to directly assess synergy.

As suggested by the Reviewer, we softened the sentence and wrote that the osimertinib-cabozantinib is additive/synergistic. However, we found that this association provoked a 90% growth inhibition in osimertinib-sensitive, which is definitely more drastic compared to what is observed with the two drugs alone (~40% inhibition for cabozantinib, ~50% for osimertinib, Fig.5A). This is confirmed by the results represented in Fig.1J, showing that the enrichment of EGFR-C797S-barcoded cells is higher with the osimertinib-cabozantinib compared to osimertinib alone. We clarified this point in the revised results.

2. *In their initial screen (Table S1) and in Figure 1K, sorafenib does not prevent the selection for Kras G12D following treatment of PC9 cells with gefitinib or the Kras G12C mutant cell line H358 with sotorasib, respectively. The authors conclude that sorafenib does not prevent the emergence of resistance in these setting. However, in Figure 2I, sorafenib does prevent the emergence of Kras G12D following treatment of the colon cancer cell line LIM1215 with cetuximab. The authors conclude that sorafenib does prevent the emergence of resistance in this setting. However, these disparate results are confusing, since in all cases resistance is driven by Kras G12D, and it would seem as though sorafenib should have similar effects. The authors should clarify this, and address whether this implies that the effect of sorafenib is dependent upon the driving oncogene, and not the resistance mechanism.*

We apologize for the misunderstanding. In our initial screen, we found that sorafenib inhibited the enrichment of KRAS-G12D cells (as shown in Table S1, sorafenib, doxorubicin and the AKT inhibitor AZD5363 were the only compounds that reduced the enrichment of KRAS-G12D cells). The fact that sorafenib could inhibit the three mechanisms of resistance modeled in our screen was, for us, one of the main reasons to consider this drug a promising hit. This inhibitory effect on the enrichment of KRAS-G12D subpopulations of EGFR-addicted cells was further confirmed in PC9 cells (Fig. 2A and Supplementary Fig. 1B), as well as in HCC827 (Supplementary Fig.2A) and H1975 (Supplementary Fig.2B) NSCLC cells and in LIM1215 CRC cells (Fig. 2H). On the contrary, we found that in KRAS-addicted H358 cells, co-treatment with sorafenib was unable to prevent the selection of KRAS-G12D cells induced by sotorasib. We agree with the Reviewer that the effects of sorafenib depend on the driving oncogene and the targeted therapy used (as discussed above). We confirmed this point by performing additional experiments using a new model of resistance to the ALK inhibitor ceritinib in a NSCLC cell line containing the EML4-ALK inversion (new Fig.1I and 5B).

3. *In Figure 3E, please label the specific Stat3 phosphorylation being measured (Y705). Given that the section preceding this was discussing the serine/threonine kinase MNK, and Stat3 can also be phosphorylated on Serine 727, it is important to be precise so that the reader does not get the impression that MNK is directly responsible for the Stat3 Y705 phosphorylation. In fact, a brief discussion of what kinases mediate Stat3 phosphorylation might be helpful.*

Agree. We have re-labeled the figures and specified in the legend that we used an antibody recognized STAT3 phosphorylated tyrosine 705.

4. *The authors state that the downregulation of Mcl1 and Myc are delayed (24 hours), but they do not show early time-points (1-2 hours) to demonstrate this.*

Agree. We performed additional time-course experiments to show that sorafenib-induced downregulation of MCL1 is delayed compared to what we observed for eIF4E and STAT3 phosphorylation (new Fig.3F). As for the effects on MYC, we considered that this was not a major point of our study and we decided not to include these results in the revised manuscript.

5. For Figure 3H, it is important to show controls that the Mnk, Stat3, and Mcl1 inhibitors are working as described (e.g. *peIF4E* and *pStat3*; showing *Mcl1* inhibition may be more difficult).

eFT-508, napabucasin and S63845 are all commercially available inhibitors that we used at the concentrations recommended by the suppliers and/or used in the literature. We have shown in several experiments that eFT-508 functions as a MNK inhibitor (Fig.3C-D, Supplementary Fig.3B and 3E-F) and we have performed an additional experiment to show that napabucasin inhibits STAT3 phosphorylation (new Supplementary Fig.4G). We also included a new Supplementary Figure (4F) to compare the effects of the three compounds alone or in combination on PC9 cells growth. However, as recognized by the Reviewer, it would be much more difficult, and we believe beyond the scope of this study, to show that S63845, a drug that was first described in a landmark Nature paper (Kotschy et al., Nature 538) and which has been tested in various clinical trials, can indeed inhibit MCL1.

6. In Figure 5A, sorafenib alone induces widespread gene expression changes. It would be important to provide a description of these changes, e.g. by performing GSEA with Hallmarks gene sets.

We agree that the transcriptional effects of sorafenib alone are interesting. KEGG pathway analysis showed that this drug affects the expression of several genes involved in important pathways, such as cell cycle, cellular senescence, DNA replication, p53 and ErbB signaling. Of note, we found a significant association between the genes downregulated by sorafenib and those involved in EGFR-tyrosine kinase inhibitor resistance (please see diagram below). However, we prefer not to include this analysis in the manuscript, since we are afraid that it would divert the focus from the main message of this section of the results, which demonstrates how co-treatment with sorafenib can inhibit the selective advantage of osimertinib-resistant cells.

KEGG pathway analysis of the genes upregulated and downregulated by sorafenib in PC9 cells.

7. The conclusion on Line 344 – 346 (These results ... suggest that, in a mixed cell population treated with both drugs, EGFR-TKI sensitive cells respond primarily to osimertinib, whereas sorafenib acts on resistant cells.) is confusing and does not seem to be supported by the data. The vast majority of cells in a bulk population of PC9 cells are sensitive to EGFR TKIs.

Therefore it is hard to understand why sorafenib would induce such widespread gene expression changes (Figure 5A, right panel) and such a strong anti-proliferative response (Figure 1I) if it were only acting on TKI-resistance cells.

We apologize for the misunderstanding, we realize that this point was not made sufficiently clear. As shown in the revised Fig.5, in the presence of co-treatment with osimertinib most of the effects of sorafenib are inhibited. This occurs only in osimertinib-sensitive cells, since, by definition, osimertinib cannot affect resistant cells. We included additional data (new Fig.5B-C) to support our model, illustrated in Supplementary Fig.8, and we clarified the related sections of the revised results and discussion.

8. How were the Osimertinib-resistant cells in Figure 5C generated? Through CRISPR-mediated mutation, or spontaneous selection?

Osimertinib resistant cells were generated by introducing the EGFR-C797S mutation through CRISPR/Cas9, followed by selection in the presence of osimertinib to eliminate the cells that didn't contain the mutation.

9. The data in Figure 6A and 6C should be quantified.

Agree. We have quantified the data from our iDISCO experiments (new Supplementary Fig.8A and D).

Reviewer 3

We thank Reviewer 3 for pointing out that some of our legends lacked important information and we apologize if we didn't sufficiently describe the number of biological replicates and independent experiments in our study. To support the reproducibility of our results, we have now included two files containing the raw data and the different independent replicates for each experiment. The manuscript has been revised in accordance with the Reviewer's specific suggestions as follows:

The authors Brunet et al. present in their current manuscript the clonal outgrowth of resistant cells in NSCLCs during EGFR-TKI resistance using DNA-barcoding and counteracting this effect by employing a combination treatment with the multikinase inhibitor sorafenib.

While an intriguing concept is presented here and the effect of the combination treatment is demonstrated in xenograft and allograft mouse models, the authors showed a severe lack of inclusion of the proper experimental controls, replicate repeats and overall reporting of "n" numbers of their experiments. Specifically this includes the representation of one experiment (of a reported 3-4 independent repeats) in the following figures: Figure 1A,B,D,E,G,K; Figure 2A,B,D,E,G,H,I; Figure 3H; Figure 4F; Extended Data Fig. 1D; Extended Data Fig. 2A. This is essentially showing an n of 1 for the majority of the data (listed above). The only figures that properly report on quantification of the data was found in extended data Fig 2B.

Other experiments presented in the figures further fail to disclose the n numbers altogether, specifically: Fig. 1H, Fig. 2C, Extended Data Fig. 1A,B,F Extended Fig. 3A,D. Extended Fig. 5G (PCR quantification), Extended Data Fig 8A, Extended Data Fig 9B. Every immunoblot shown, additionally, has no reporting on the experimental n numbers at all in the respective figures. The following blots are missing the proper loading controls: Figure 3A-C, Extended Data Fig 3B and 3F. Figure 4A, Extended Figure 4A first blot, also the second and third do not report on total STAT3, Extended Data Figure 4B.

Furthermore, none of the colony formation assays are quantified nor an “n” number mentioned in the figure captions, namely: Figure 1F, 1I and 1L, Extended data Fig 1C & E, Extended data Fig 2 C+F, Extended data Fig 4E, some of which also seem to be acquired out of focus with phase contrast artefacts in the images, Extended Data Fig. 9A.

Therefore little to no conclusions can be drawn from a majority of the figures about the validity or reproducibility of the reported findings in the manuscript in its present form. In order to make the data presented in the manuscript interpretable, authors would be requested to show the following: (1) individual data points in the bar graphs as well as the combination of the independent experimental repeats in these graphs, and NOT an n of 1 of a representative repeat (specific figures listed above). Additionally SEMs should be plotted as bot +/- in the graphs. (2) n numbers for the data presented as reported above, where there is currently no mention of them at all. (3) the appropriate loading controls of the immunoblots as well as annotations of ladder kDAs for each blot. (4) raw data of the immunoblots and their repeats. If there are no repeats of the immunoblots these will need to be performed and submitted. (5) quantifications of the colony formation assays, as well as again the n number repeats of these experiments.

We are sorry for the misunderstanding. We agree that some legends should have described more clearly the number of independent experiments and we apologize if the representation of our results was not sufficiently clear. However, we believe the Reviewer was overly harsh when he/she wrote that our manuscript severely lacked proper experimental controls and replicate repeats. Our barcoding experiments contained several biological replicates per condition, *i.e.* different flasks of cells (generally 4, in the revised manuscript the legends indicate the exact number of biological replicates in each experiment). Our histograms showed the mean \pm SEM of these biological replicates for each condition. As suggested by the Reviewer, in the revised manuscript we have represented each individual biological replicate, which, we agree, is now considered a more appropriate way to depict the results. Each experiment has been independently performed at least three times (this is now indicated for each experiment in the legends of the revised manuscript) and we did statistical analysis for each of them. We prefer to show representative experiments (which include several biological replicates per condition), and by representative, we mean that similar statistically significant differences were observed in each experiment. As requested by the Reviewer, we have submitted a file containing all the raw data of each experiment, as well as one other file containing all the replicates for each panel of the revised manuscript. We believe this additional information will conclusively demonstrate the reproducibility of our results.

The reviewer wrote that some of our immunoblots lacked loading controls, such as for example in Figures 3A-C, in which we displayed the effects of treatment on phospho-eIF4E and phospho-ERK. However, these figures show the levels of the total form of these proteins, which is the gold-standard control for studies on phosphorylated proteins. We recognize that few immunoblots with phospho-antibodies didn't display the corresponding total proteins and we rectified these figures in the revised manuscript.

The Reviewer also thought that our Extended Data Fig. 9A (now revised Supplementary Fig.11A) contained phase contrast artefacts. We were surprised by this statement, since the panel was generated by simply taking photographs of the whole 6-well plates, as we did for all our colony formation assays (please see “Brunet_raw data” file). The figure below shows that this panel clearly does not contain any artefact.

Raw images of the colony formation assays used to generate Supplementary Fig.11A (shown on the left).

In summary, in response to the Reviewer general comments, (1) we showed individual data points for each biological replicates and included the raw data and the independent replicate experiments in the two submitted files “Brunet_raw data” and “Brunet_experiment replicates”; (2) for each panel, we have indicated the number of biological replicates and independent experiments in the legends; (3/4) we have verified that immunoblots contained appropriate loading controls, we have included size annotations and we have submitted files with the raw data of the immunoblots and their repeats; (5) we have submitted the photos of the whole plates for each colony formation assays, as well those of all the different independent experiments.

Our responses to the other specific points of the Reviewer are indicated below:

Figure 6: Why were the Ctr mice sacrificed at 200mm² and the treatment mice after 4 weeks of treatment (i.e. at 800mm²)? In order to be truly comparable, mice should have been treated for the same time with vehicles and/or sacrificed when reaching a certain tumour size, i.e. 800 mm² for all experimental groups, not just the treated animals. There can be no reliable comparison of clonal size outgrowth to the control if it was already sacrificed at smaller volume/earlier and no conclusions drawn with regards to the increase in mutational clonal sizes if the controls are not allowed to reach the same size as the treated tumours. As a side note, the halving of the Osimertinib tumours, seems to be more a motivation to make the imaging more feasible than anything else, since no quantifications are shown (see minor comment below).

We certainly agree that it would have been unfair to compare tumors of different size, but we don't understand why this Reviewer assumes that in our experiments the tumors from treated mice were much larger than those from control mice. Fig.5A and 5C, as well as the quantification illustrated in the new Supplementary Fig.9A and 9B, show that the size of the control and treated tumors were actually comparable. Since both osimertinib and the osimertinib-sorafenib combination have a strong inhibitory effect on tumor growth (Fig. 6D and 7A), it would have been unfeasible to sacrifice the mice at the same time, since the control tumors would have been enormous compared to the treated ones. Also, we preferred not to sacrifice the mice when their tumors reached a certain volume, because in this case the duration of the treatment wouldn't have been the same for the different conditions (osimertinib vs combination) and replicates. Therefore, we decided to sacrifice the control mice at time zero, corresponding to the beginning of the treatment, while treated mice were

sacrificed after four weeks of treatment, which, as we knew from previous experiments, resulted in tumors of similar sizes for the three conditions.

Figure 6D: An additive/cumulative tumour volume of both implanted flank tumours is plotted here for each animal or the volume of the larger of the two implanted tumours?

As shown in the “Brunet_raw data” file, all tumors were included to draw our plots.

Minor comments:

Figure 2A: Why was the combination treatment of Osimertinib + Trametinib shown only for the PIK3CA-E545K mutant and not for the others?

The goal of our study was not to compare the effects of the osimertinib-sorafenib and the osimertinib-trametinib combinations. However, since trametinib could inhibit the enrichment of resistant cells containing certain mutations, such as secondary EGFR mutations (Guernet et al., Mol. Cell 63 and Table S1), we wanted to show that this MEK inhibitor was not efficient against other mechanism of resistance, including PIK3CA-E545K. Similarly, in some of the other graphs in Fig.2 we included the MET/ALK inhibitor crizotinib as a positive control (Fig.2C-D).

Page 13, line 312-314: The authors state: “[...] while other multikinase inhibitors, such as sunitinib and regorafenib, failed to effectively down-regulate the receptor (Extended Data Fig.5G).” This statement needs to be softened, since looking at the immunoblots, 5 μ M regorafenib and sunitinib in the subsequent blot clearly do reduce the expression levels of the EGFR. Again, repeats of these experiments and quantifications would benefit the interpretation of this data.

We have toned down this sentence.

Page 13, line 314, typo “didi” – did

Agree. We have corrected the typo.

Page 13, line 314-315: This statement: “Of note, we didi not observe any effect of sorafenib on EGFR mRNA levels (Extended Data Fig.5G), [...]” cannot be made without any statistical analysis shown.

Agree. EGFR mRNA levels in the presence of sorafenib are not statistically different compared to the control, as it is now indicated in the figure (revised Supplementary Fig.5C).

Figure 6A: The authors are asked to quantify the regions of the individual labelled mutations in order to make the claims these clones were enriched in the Osimertinib treated samples.

Agree. We have quantified both Fig. 6A and 6C and represented the data in the new Supplementary Fig.9A and 9D, respectively.

Figure 6A,C: The authors are requested to include the scale bar measurement in the figure legend.

Agree. We have included the scale bar measurement in the legend.

Figure 6E, The authors are requested to plot any censored events in the Kaplan-Meier curve, as well as include relevant statistics in the figure, e.g. a log rank test between experimental groups.

The number of censored events is very limited (3 out of 29 mice) and it can now be easily found in the Brunet_raw data file, so we don't think it would be very useful to also include it in the plot, also considering it is a mouse experiment. We have included statistics in the figures.

Page 15 line 383-384: This statement: "The PDC line YUX-1024 forms tumors that are histopathologically similar to patient derived xenografts obtained from the same patient (Extended Data Fig.7B)." is just not true when examining the presented H&Es images, with the PDX F1 & 2 showing significantly enlarged nuclei and stronger stromal features compared to the PDC tumour, and this statement therefore needs to be softened.

We have toned down the sentence.

Figure 7D, Please include scale bars in the figure.

We have included the scale bars in the figure.

Figure 7F, The authors are requested to plot any censored events in the Kaplan-Meier curve, as well as include relevant statistics in the figure, e.g. a log rank test between experimental groups.

Agree. Please see our response to the similar point raised for Fig.6E.

Extended Data Fig.9E: This quantification should be moved to the main Figure 7 and shown with the necessary statistics in order to draw conclusions about the increase in certain immune populations.

We have modified the Figure, which now corresponds to Supplementary Fig. 11E, and included the statistical analysis, but we'd prefer to keep it in the supplementary.

Point-by-point response to the Reviewers

We thank the Reviewers for their positive evaluation of our revised manuscript. Our responses to their comments are indicated below:

Reviewer 1:

The authors have provided additional data and addressed some of the reviewer's questions. Studies using sorafenib including clinical trials were conducted more than ten years ago, and I certainly agree with their argument that previous clinical trials on the combination of EGFR-TKIs and sorafenib included only a small number of patients. However, the authors have not provided convincing results to overturn the findings that sorafenib may not have a major impact on the emergence of resistance to EGFR-TKIs. Another concern is that the manuscript does not provide mechanistic insights regarding effects of sorafenib on emergence of resistance to EGFR-TKIs. While they conclude that sorafenib prevents the emergence of EGFR-TKI resistant cells through the inhibition of MNK, STAT3, eIF4, and MCL1, each mechanism lacks in-depth investigation and the manuscript seems unfocused.

We appreciate Reviewer 1's constructive comments. Our study shows that sorafenib delays the acquisition of resistance to EGFR-TKIs through a mechanism that involves inhibition of MNK-induced phosphorylation of eIF4E, STAT3 phosphorylation, MCL1 expression and EGFR protein stability. More specifically, we demonstrated that this drug not only directly inhibits the catalytic activity of MNKs (Fig. 3A-D), but it also promotes the degradation of MCL1 and EGFR, through the proteasome and lysosome pathways, respectively (Fig. 3H-I & 4D-G). Consistent with these findings, we used a pharmacological approach to show that the effects of sorafenib can be, on the one hand, mimicked by concurrent inhibition of MNK, STAT3 and MCL1 (Fig. 3J), and, on the other hand, rescued by prevention of lysosomal degradation (Fig. 4G). In response to the Reviewer's comments, we have included in this second revision new data from knockdown and overexpression experiments, which confirm this mechanism, as detailed below.

As for the comparison between our present findings and previous clinical reports on the combination of sorafenib and 1st generation EGFR-TKIs, the Reviewer agrees that such studies were very limited. As described in the discussion, two of the three papers available, based on only 8 and 7 patients, respectively, did not include a EGFR-TKI monotherapy arm, making it impossible to assess the potential benefit of the combination. While it compared the effects of the combination with those of EGFR-TKI alone, the third study is based on an even smaller number of patients (two patients for the combination, three for EGFR-TKI). These reports can hardly be considered convincing evidence that need to be overturned, and it seems fair to conclude that the clinical benefits of the combination were never properly tested. In our study we used different *in vitro* and *in vivo* approaches to show that sorafenib can delay the emergence of resistance to EGFR-TKIs. The large amount of preclinical data described in the present manuscript will provide the rationale to support a clinical trial to conclusively assess the efficacy of this combination in NSCLC patients.

Major

Fig. 1F-I and Fig. 2: The authors showed that sorafenib can effectively prevent the emergence of resistance in cancer with EGFR mutation, but not in cells addicted to other oncogenic alterations such as KRAS and ALK in Fig. 1F-I. However, Fig. 2 shows that sorafenib is also effective against cells with resistance mechanisms independent on EGFR pathway signaling, including KRAS-G12D, BRAF-V600E, and EML4-ALK, in PC9 cells. It is not clear why sorafenib is effective in EGFR-mutated lung cancer with various resistance

mechanisms independent of EGFR pathway signaling, despite poor efficacy against KRAS G12C or ALK fusion cell lines.

We showed that CRISPR/Cas9-induced mutation of KRAS or ALK can confer a selective advantage in the presence of osimertinib to cancer cells addicted to mutant EGFR. However, this doesn't mean that these cells have become addicted to KRAS or ALK, and behave like KRAS- or ALK-driven cancer cells. EGFR, KRAS and ALK induce different oncogenic programs in NSCLC and tumor cells cannot be reprogrammed by the simple acquisition of a new mutation. Indeed, although some cases have been reported in the literature, KRAS and especially ALK mutations are relatively uncommon mechanisms of resistance to EGFR-TKIs, and it is widely recognized that EGFR, KRAS and ALK mutations are almost always mutually exclusive in NSCLC. We have modified the text to clarify this point.

Fig. 1L and Fig. 5A: The authors showed that sorafenib alone does not affect cell survival in PC9 cells in Fig. 1L (page 9, line 195). However, Fig. 5A indicates that cell viability in PC9 parental cells after treatment with sorafenib alone is around 50%. Where does this discrepancy come from? Furthermore, cell viability in OR-PC9 cells after treatment with sorafenib alone was also around 50%. Despite PC9 parental cells having far fewer osimertinib-resistant clones than OE-PC9 cells, how are these results similar?

We showed that sorafenib inhibits proliferation (Fig. 1K), but does not induce apoptosis (Fig. 1L) in PC9 cells. In the viability assay shown in Figure 5A, the cells were treated in the presence or the absence of different drugs for 5 days, and the number of viable cells was assessed by Cell Titer Glow. The figure shows less cells in the sorafenib group, which is consistent with the inhibition of cell proliferation induced by this compound. As for the similar effects of sorafenib on osimertinib sensitive and resistant cells, this is consistent with our model. Our data show that sorafenib, as a single agent, exerts the same effects in osimertinib sensitive and resistant cells. These effects are also observed with the combination in osimertinib-resistant cells, since these cells are, by definition, insensitive to osimertinib and they thus only respond to sorafenib. On the contrary, in osimertinib-sensitive cells treated with the combination, osimertinib impairs the effects of sorafenib, as shown by the lack of additive effects by the two drugs and confirmed by our mechanistic studies. This important point has been discussed in details in the manuscript, including a diagram (Supplementary Figure 8) to illustrate the model.

Fig.3: The authors conclude that sorafenib prevents the emergence of EGFR-TKI resistant cells through the inhibition of MNK, STAT3, and MCL1. However, no functional studies such as knockdown/knockout and overexpression have been conducted. Therefore, it remains unclear whether these inhibitions directly prevent the emergence of EGFR-TKI-resistant clones.

Agree. To address this point, we have generated three different lentiviral constructs for inducible expression of a dominant negative MNK1 (DN-MNK1) and shRNAs targeting either MCL1 or STAT3. As shown in the new Figure 3K, even a moderate inhibition of the expression of MCL1 and STAT3 and the phosphorylation of eIF4E (induced by DN-MNK1) can significantly reduce the emergence of osimertinib-resistant cells. These data are consistent with our previous experiments based on the pharmacological inhibition of MNK, STAT3 and MCL1 (Fig. 3J) and further demonstrate the role of these three factors in the mechanism of sorafenib in NSCLC cells.

Fig. 3H: The reviewer does not understand what the authors question here. MCL1 downregulation was observed by CHX due to its inhibition of protein synthesis, and so did co-treatment with sorafenib. In this situation, the experiment does NOT imply that "sorafenib

does not affect MCL1 protein synthesis” because MCL1 is already inhibited by CHX to begin with. Or do the authors mean that sorafenib may enhance downregulation of MCL1?

We are sorry for the misunderstanding. The experiment depicted in Fig. 3H shows that sorafenib can still inhibit MCL1 levels when the synthesis of new proteins is blocked by CHX, indicating that sorafenib affects MCL1 at a post-translational level. We have modified the text to clarify this point.

Fig. 8: In response to my comments regarding drug-tolerant persister cells, the authors evaluated the long-term effects in combination therapy in vitro and in vivo. However, they only evaluated PC9 cells which premixed with parental and DTEP cells in vitro. The reviewer is concerned that this is too artificial and not enough rationale for the 1:100 premix. No evaluation of characteristics of DTEP cells (e.g. sensitivity to osimertinib, which should be reversed) after labeling is provided. In vivo evaluation of lung cancer cells is all based on PC9 cell line (BEM cells are fibroblasts). Lastly, adding other parameters than hematopoietic measurements for toxicity of the combination therapy would be beneficial.

In the revised manuscript, we used PC9 cells and BEM4 cells to investigate the effects of sorafenib on persister cells *in vitro* and *in vivo*, respectively. For the *in vitro* experiments with PC9 cells we used two different approaches. One was based on long term monitoring of drug response through the incucyte imaging system, which is a gold standard strategy to investigate the emergence of tolerant/persister cells. The second complementary approach was based on PC9 DTEP cells, which were characterized in depth several years ago in a seminal Cell paper by the Settleman lab, one of the very first studies describing tolerant/persister cells. We added a new Supplementary Figure (Supplementary Fig. 12A) to illustrate the different sensitivity to osimertinib of parental, fully resistant and DTEP PC9 cells. In the experiment described in Figure 8B, we mixed parental and genetically labeled DTEP cells to generate a mass population containing both sensitive and persister cells and simultaneously compare their response to osimertinib and the osimertinib-sorafenib combination. To analyze the proportion of barcoded cells in each biological replicate, we performed qPCR from 100 ng of genomic DNA, which correspond roughly to 15 000 cells. The 1 to 100 ratio used is a compromise that enables comfortable detection of DTEP cells in control conditions, as well as accurate measurement of the enrichment induced by osimertinib. Of note, the DTEP experiment was already included in the first version of our manuscript.

In our study, the osimertinib-sorafenib combination was tested in two different types of human NSCLC cells, PC9 and the PDC line YUX-1024 (Fig. 6 and Fig. 7A-B). To investigate the effects of sorafenib on persister cells *in vivo*, we used BEM4 cells, which are not derived from NSCLC, as pointed out by the Reviewer, but they are cells addicted to mutant EGFR that can form tumors in immunocompetent mice. We decided to generate this new model for two reasons. First, several evidence indicate that the immune system could play a role in the response to EGFR-TKIs. Second, murine NSCLC cells addicted to mutant EGFR are not available. We thank the Reviewer for his/her last comment related to the *in vivo* toxicity of the combination. We included a new supplementary figure (Supplementary Fig. 13) in the revised manuscript to depict the weight of the mice during the treatment in our five different *in vivo* experiments. These data are consistent with the blood parameters shown in Fig. 8H and support the low mouse toxicity of the combination.

Reviewer 2:

In the revised manuscript the authors address the majority of the issues raised in the initial review, and as a result the manuscript was significantly improved. I do not have any further concerns.

We greatly appreciate Reviewer 2's very positive comments.

Reviewer 3:

The manuscript by Brunet et al. has greatly improved post rebuttal and the authors have gone to great length to show their work in the detail that is required for a reader audience to interpret the presented data fully. This reviewer is happy with the authors responses, having addressed the majority of this reviewers comments fully and using appropriate data presentation in their manuscript.

We greatly appreciate Reviewer 3's very positive comments.